# Towards Gradient-based Bilevel Optimization with Non-convex Followers and Beyond

**Risheng Liu**[1,2]    **Yaohua Liu**[1]    **Shangzhi Zeng**[3]    **Jin Zhang** [*4,5]

[1]International School of Information Science & Engineering, DUT    [2]Pazhou Lab, Guangzhou
[3]Department of Mathematics and Statistics, UVic
[4]Department of Mathematics, SUSTech    [5]National Center for Applied Mathematics Shenzhen
rsliu@dlut.edu.cn  liuyaohua_918@163.com
zengshangzhi@gmail.com  zhangj9@sustech.edu.cn

## Abstract

In recent years, Bi-Level Optimization (BLO) techniques have received extensive attentions from both learning and vision communities. A variety of BLO models in complex and practical tasks are of non-convex follower structure in nature (a.k.a., without Lower-Level Convexity, LLC for short). However, this challenging class of BLOs is lack of developments on both efficient solution strategies and solid theoretical guarantees. In this work, we propose a new algorithmic framework, named Initialization Auxiliary and Pessimistic Trajectory Truncated Gradient Method (IAPTT-GM), to partially address the above issues. In particular, by introducing an auxiliary as initialization to guide the optimization dynamics and designing a pessimistic trajectory truncation operation, we construct a reliable approximate version of the original BLO in the absence of LLC hypothesis. Our theoretical investigations establish the convergence of solutions returned by IAPTT-GM towards those of the original BLO without LLC. As an additional bonus, we also theoretically justify the quality of our IAPTT-GM embedded with Nesterov's accelerated dynamics under LLC. The experimental results confirm both the convergence of our algorithm without LLC, and the theoretical findings under LLC.

## 1 Introduction

Bi-Level Optimization (BLO) has been widely used to formulate problems in the field of deep learning [1, 2], especially for hyperparameter optimization [3, 4, 5], meta learning [6, 7, 8, 9], neural architecture search [10, 11, 12], adversarial learning [13], and reinforcement learning [14], etc. BLO aims to tackle nested optimization structures appearing in applications, which has emerged as a prevailing optimization technique for modern machine learning tasks with underlying hierarchy. In the last decade, a large number of BLO methods have been proposed to address different machine learning tasks. In fact, Gradient Methods (GMs), which can effectively handle BLO problems of large scale, thus gain popularity. According to different types of strategies for gradient calculations, existing GMs can be divided into two categories, i.e., the explicit approaches which aim to replace the Lower-Level (LL) problem with dynamic iterations and implicit schemes that apply the implicit function theorem to formulate the first-order optimality condition of the LL problem.

**Explicit Gradient Methods for BLOs.**    In this type, solving the LL problem is regarded as the evolution path of the dynamic system starting from a given initial point of the LL variable. The gradient of the Upper-Level (UL) variable can be directly calculated by automatic differentiation

---

*Corresponding author

35th Conference on Neural Information Processing Systems (NeurIPS 2021).

based on the trajectory of LL variable. This class of methods can be further divided into three types, namely, recurrence-based EG (e.g., [3, 15, 6, 16, 10]), initialization-based EG (e.g., [17, 18]) and proxy-based EG methods (e.g., [19, 20, 21, 22]), differing from each other in the way of accessing the gradient of constructed dynamic trajectory. While most of this type of works assume the LLC and Lower-Level Singleton (LLS) to simplify their optimization process and theoretical analysis, cases where the LLS assumption does not hold have been tackled in the recent work [23, 24]. In particular, to eliminate the LLS assumption which is too restrctive to be satisfied in real-world complex tasks, [23] first considers incorporating UL objective information into the dynamic iterations, but the more general cases where LLC does not hold remain unsolved. On the other hand, while most of the mentioned works focus on the asymptotic convergence, the progress on nonasymptotic convergence analysis has been recently witnessed see, e.g., [25, 26, 27].

**Implicit Gradient Methods for BLOs.**   This type, also known as implicit differentiation [28, 7, 29], replaces the LL problem with its first-order optimality condition and uses the implicit function theorem to calculate the gradient of the UL problem by solving a linear system. This method decouples the calculation of UL gradient from the dynamic system of LL, resulting in a significant speed increase when the dynamic system iterates many times. However, because of the burden originated from computing a Hessian matrix and its inverse, IG methods are usually computationally expensive when linear systems are ill-conditioned. To alleviate this computational issue, there are mainly two kinds of techniques, i.e., IG based on Linear System (LS) [28, 7] and Neumann Series (NS) [29]. On the theoretical side, IG methods rely on the strong convexity of LL problems heavily, which is even more restrictive than the LLC and LLS together.

**Initialization Optimization for Learning.**   In deep learning, the selection of initialization scheme has a great influence on the training speed and performance [30]. As the most representative work in recent years, Model-Agnostic Meta-Learning (MAML) [31] applies the same initialization to all tasks, and is optimized by a loss function common to the task that evaluates the effect of the initial value, resulting in an initialization that achieves good generalization performance with only a few gradient steps on new tasks. Due to its simple form this method has been widely studied and applied [32, 33, 34, 35]. [36] noticed that not all network parameters are suitable for the same initialization, and therefore proposed a strategy to apply co-initialization only on a part of parameters. In theory, [37, 38, 39] give comprehensive study on the convergence and convergence rate of MAML and some MAML-type approaches based on the meta objective function. However, the convergence theory of these existing results are given based on the loss function for evaluating initial values, and the convergence analysis of such type of methods from the perspective of each task is still lacked.

**Value-Function Approach.**   The value function based methods have also emerged as a promising branch to solve BLO problems [40]. Under the special case where the LL is jointly convex with respect to both the UL and LL variables, the BLO problem can be equivalently reformulated into a difference-of-convex program [41], which is numerically solvable. Typically, by reformulating the BLO into an Inner Single Bi-level (ISB) optimization problem with value-function approach, a gradient-based interior-point method name BVFIM [42] is proposed to solve the BLO tasks, which effectively avoids the expensive Hessian-vector and Jacobian-vector products. Generally speaking, the value-function does not admit an explicit form, and is always nonsmooth, non-convex and with jumps.

## 1.1   Our Motivations and Contributions

As mentioned above, some theoretical progresses have long been witnessed in diversified learning areas, but for most existing BLO methods, extra restrictive assumptions (e.g., LLS, LLC and LL strong convexity) have to be enforced. Their algorithm design and associated theoretical analysis actually are only valid for optimization with a simplified problem structure. Unfortunately, it has been well recognized that LL non-convexity frequently appears in a variety of applications, e.g., sparse $\ell_q$ regularization ($0 < q < 1$) for avoiding over-fitting, and learning parameters of coupled multi-layer neural networks, etc. Therefore, in challenging real-world scenarios, we are usually required to consider BLO problems where *these assumptions (e.g., LLS, LLC and even LL strong convexity) are naturally violated*. These fundamental theoretical issues motivate us to propose a series of new techniques to address BLO with non-convex LL problems, which have been frequently appeared in various learning applications.

In particular, by introducing an Initialization Auxiliary (IA) to the LL optimization dynamics and operating a Pessimistic Trajectory Truncation (PTT) strategy during the UL approximation, we construct a Gradient-based Method (GM), named IAPTT-GM, to address BLO in challenging optimization scenarios (i.e., with non-convex follower tasks). We analyze the convergence behaviors of IAPTT-GM on BLOs without LLC and also investigate theoretical properties of the accelerated version of our algorithm on BLOs under LLC. Extensive experiments verify our theoretical results and demonstrate the effectiveness of IAPTT-GM on different learning applications. The main contributions of our IAPTT-GM are summarized as follows:

- We propose IA and PTT, two new mechanisms to efficiently handle complex BLOs where the follower is facing with a non-convex task (i.e., without LLS and even LLC). IA actually paves the way for jointly optimizing both the UL variables and the dynamical initialization, while PTT adaptively reduces the complexity of backward recurrent propagation.

- To our best knowledge, we establish the first strict convergence guarantee for gradient-based method on BLOs with non-convex follower tasks. We also justify the quality of our IAPTT-GM embedded with Nesterov's accelerated dynamics under LLC.

- We conduct a series of experiments to verify our theoretical findings and evaluate IAPTT-GM on various challenging BLOs, in which the follower tasks are either with non-convex loss functions (e.g., few-shot learning) or coupled network structures (e.g., data hyper-cleaning).

## 2   The Proposed Algorithmic Framework

In this work, we consider the BLO problem in the form:

$$\min_{\mathbf{x} \in \mathcal{X}, \mathbf{y}} F(\mathbf{x}, \mathbf{y}), \quad s.t. \quad \mathbf{y} \in \mathcal{S}(\mathbf{x}), \tag{1}$$

where $\mathbf{x} \in \mathbb{R}^n, \mathbf{y} \in \mathbb{R}^m$ are UL and LL variables respectively, and $\mathcal{S}(\mathbf{x})$ denotes the set of solutions of the LL problem, i.e.,

$$\mathcal{S}(\mathbf{x}) := \arg \min_{\mathbf{y} \in \mathcal{Y}} f(\mathbf{x}, \mathbf{y}), \tag{2}$$

where $f$ is differentiable w.r.t. $\mathbf{y}$. To ensure the BLO model in Eq. (1) is well-defined, we assume that $\mathcal{S}(\mathbf{x})$ is nonempty for all $\mathbf{x} \in \mathcal{X}$. Observe further that the above BLO is structurally different from those in existing literature in the sense that no convexity assumption is required in the LL problem.

In the following, we describe the proposed IAPTT-GM to solve the class of BLOs defined in Eq. (1). The mechanism of a classical dynamics-embedded gradient method, approximates the LL solution via a dynamical system drawn from optimization iterations. Choosing gradient descent as the optimization dynamics for example, the approximation $\mathbf{y}_K(\mathbf{x})$ is accessed by operations repeatedly performed by $K - 1$ steps parameterized by UL variable $\mathbf{x}$

$$\mathbf{y}_{k+1}(\mathbf{x}) = \mathbf{y}_k(\mathbf{x}) - s\nabla_{\mathbf{y}} f(\mathbf{x}, \mathbf{y}_k(\mathbf{x})), \ k = 0, \cdots, K - 1, \tag{3}$$

where $s$ is a step size, and $\mathbf{y}_0$ is a fixed initial value.

### 2.1   Initialization Auxiliary

Embedding the dynamical iterations into the UL problem returns the approximate version $F(\mathbf{x}, \mathbf{y}_K(\mathbf{x}))$. As long as $\nabla_{\mathbf{y}} f(\mathbf{x}, \mathbf{y}_K(\mathbf{x}))$ uniformly converges to zero w.r.t. UL variable $\mathbf{x}$ varying in $\mathcal{X}$, we call this a good approximation. To this end, usually restrictive LL strong convexity assumptions are imposed, thus the desired convergence of solutions of approximation problems towards those of the original BLO follows. By drawing inspiration from the classic dynamics-embedded gradient method which replaces the LL problem with certain optimization dynamics, hence resulting in an approximation of the bi-level problem, we propose a new gradient scheme to solve BLO in Eq. (1) without LLC restriction. Specifically, we let $K$ be a prescribed positive integer and construct the following approximation $\mathbf{y}_K(\mathbf{x}, \mathbf{z})$ of the LL solution drawn from projected gradient descent iterations

$$\begin{aligned} \mathbf{y}_0(\mathbf{x}, \mathbf{z}) &= \mathbf{z}, \\ \mathbf{y}_{k+1}(\mathbf{x}, \mathbf{z}) &= \texttt{Proj}_{\mathcal{Y}}(\mathbf{y}_k(\mathbf{x}, \mathbf{z}) - \alpha_{\mathbf{y}}^k \nabla_{\mathbf{y}} f(\mathbf{x}, \mathbf{y}_k(\mathbf{x}, \mathbf{z}))), \ k = 0, \cdots, K - 1, \end{aligned} \tag{4}$$

where $\{\alpha_{\mathbf{y}}^k\}$ is a sequence of steps sizes. We next embed the dynamical iterations $\mathbf{y}_k(\mathbf{x}, \mathbf{z})$ into $\max_{1 \le k \le K} \{F(\mathbf{x}, \mathbf{y}_k(\mathbf{x}, \mathbf{z}))\}$, which can be regarded as a pessimistic trajectory truncation of the UL objective. The mechanism of the above scheme, in comparison, accesses the approximation parameterized by UL variable $\mathbf{x}$ and LL initial point $\mathbf{y}_0$. Our motivation for the initialization auxiliary variable $\mathbf{z}$ comes from convergence theory [42] of non-convex first-order optimization methods. In fact, when the non-convex LL problem admits multiple solutions, the gradient descent steps with a "bad" initial point $\mathbf{y}_0$ cannot return a desired point in the LL solution set, simultaneously optimizing the UL objective. To overcome such a difficulty, instead of using a fixed initial value, we introduce an initialization auxiliary variable $\mathbf{z}$. Therefore, when it comes to solving the UL approximation problems, together with the UL variable $\mathbf{x}$, the auxiliary variable $\mathbf{z}$ is also updated and hence optimized. As a consequence, we may search for the "best" initial value, starting from which the gradient descent steps approach a solution to the BLO in Eq. (1), i.e., a point in the LL solution set, simultaneously minimizing the UL objective.

## 2.2 Pessimistic Trajectory Truncation

This is a striking feature of our algorithm that significantly differs from existing methods and leads to some new convergence results without LLC. The motivation for the design of pessimistic trajectory truncation comes again from convergence theory of non-convex first-order optimization methods. It is understood that when LL is non-convex, $\mathcal{R}_\alpha(\mathbf{x}, \mathbf{y}_K(\mathbf{x}, \mathbf{z}))$ may not uniformly converge w.r.t. $\mathbf{x}$ and $\mathbf{z}$, where $\mathcal{R}_\alpha(\mathbf{x}, \mathbf{y})$ is the proximal gradient residual mapping defined as $\mathcal{R}_\alpha(\mathbf{x}, \mathbf{y}) := \mathbf{y} - \mathrm{Proj}_{\mathcal{Y}}(\mathbf{y} - \alpha \nabla_{\mathbf{y}} f(\mathbf{x}, \mathbf{y}))$ [1], which can be used as a measurement of the optimality of LL problem in Eq. (2). Thus a direct embedding of $\mathbf{y}_K(\mathbf{x}, \mathbf{z})$ into UL objective $F(\mathbf{x}, \mathbf{y})$ may not necessarily provide an appropriate approximation. Fortunately, it is also understood that, for each $\mathbf{x} \in \mathcal{X}$, $\mathbf{z} \in \mathcal{Y}$ and $K > 0$, there exists at

---

**Algorithm 1** The Proposed IAPTT-GM

1: Initialize $\mathbf{x}^0$ and $\mathbf{z}^0$.
2: **for** $t = 0 \to T - 1$ **do**
3:     $\mathbf{y}_0 = \mathbf{z}^t$.
4:     **for** $k = 0 \to K - 1$ **do**
5:         % LL Updating with $\mathbf{x}^t$ and $\mathbf{z}^t$
6:         $\mathbf{y}_{k+1} = \mathrm{Proj}_{\mathcal{Y}}(\mathbf{y}_k - \alpha_{\mathbf{y}}^k \nabla_{\mathbf{y}} f(\mathbf{x}^t, \mathbf{y}_k))$.
7:     **end for**
8:     % Pessimistic Trajectory Truncation
9:     $\bar{k} = \arg\max_k \{F(\mathbf{x}, \mathbf{y}_k)\}_{k=1}^K$.
10:    % UL Updating with $\mathbf{y}_{\bar{k}}(\mathbf{x}, \mathbf{z})$
11:    $\mathbf{x}^{t+1} = \mathrm{Proj}_{\mathcal{X}}(\mathbf{x}^t - \alpha_{\mathbf{x}} \nabla_{\mathbf{x}} F(\mathbf{x}^t, \mathbf{y}_{\bar{k}}))$.
12:    % Initialization Updating with $\mathbf{y}_{\bar{k}}(\mathbf{x}, \mathbf{z})$
13:    $\mathbf{z}^{t+1} = \mathrm{Proj}_{\mathcal{Y}}(\mathbf{z}^t - \alpha_{\mathbf{z}} \nabla_{\mathbf{z}} F(\mathbf{x}^t, \mathbf{y}_{\bar{k}}))$.
14: **end for**

---

least a $\tilde{K}$ such that along the selection $\mathbf{y}_{\tilde{K}}(\mathbf{x}, \mathbf{z})$, $\mathcal{R}_\alpha(\mathbf{x}, \mathbf{y}_{\tilde{K}}(\mathbf{x}, \mathbf{z}))$ uniformly converges to zero w.r.t. $\mathbf{x}$ and $\mathbf{z}$, as $K$ tending infinity.

However, in general, it is too ambitious to expect an explicit identification of the exact selection $\mathbf{y}_{\tilde{K}}(\mathbf{x}, \mathbf{z})$. Alternatively, we consider a pessimistic strategy, minimizing the worst case of all selections of $\{\mathbf{y}_k(\mathbf{x}, \mathbf{z})\}$, i.e., $\max_{1 \le k \le K} \{F(\mathbf{x}, \mathbf{y}_k(\mathbf{x}, \mathbf{z}))\}$. By doing so, we successfully reach a good approximation. In addition to the theoretical convergence, we also benefit from this pessimistic strategy in a numerical sense. The pessimistic $\max$ operation always results in a favorable trajectory truncation smaller than $K$. Consequently, this technique offers inexpensive computational cost for computing the hyper-gradient through back propagation, as shown in the numerical experiments.

To conclude this section, we state the complete IAPTT-GM in Algorithm 1. Note that $K$ and $T$ represent the numbers of inner and outer iterations, respectively.

## 3 Theoretical Investigations

With the purpose of studying the convergence of dynamics-embedded gradient method for BLO without LLC, we involve two signature features in our algorithmic design, i.e., initialization auxiliary and pessimistic trajectory truncated. This section is devoted to the convergence analysis of our proposed algorithm with and without LLC assumption. Please notice that all the proofs of our theoretical results are stated in the Supplemental Material.

---

[1] $\mathcal{R}_\alpha(\mathbf{x}, \mathbf{y})$ reduces to $\alpha \nabla_{\mathbf{y}} f(\mathbf{x}, \mathbf{y})$ when $\mathcal{Y}$ is taken as $\mathbb{R}^m$.

## 3.1 Convergence Analysis of IAPTT-GM for BLO with Non-convex Followers

In this part, we conduct the convergence analysis of the IAPTT-GM for solving BLO in Eq. (1) without LLC. Before presenting our main convergence results, we introduce some notations related to BLO. With introduced function $\varphi(\mathbf{x}) := \inf_{\mathbf{y} \in \mathcal{S}(\mathbf{x})} F(\mathbf{x}, \mathbf{y})$, the BLO in Eq. (1) can be rewritten as

$$\min_{\mathbf{x} \in \mathcal{X}} \varphi(\mathbf{x}). \tag{5}$$

With given $K \geq 1$ and defining $\varphi_K(\mathbf{x}, \mathbf{z}) := \max_k \{F(\mathbf{x}, \mathbf{y}_k(\mathbf{x}, \mathbf{z}))\}$ with $\{\mathbf{y}_k(\mathbf{x}, \mathbf{z})\}$ defined in Eq. (4), our proposed IAPTT-GM generates sequence $\{(\mathbf{x}^t, \mathbf{z}^t)\}$ for solving following approximation problem to BLO in Eq. (5),

$$\min_{\mathbf{x} \in \mathcal{X}, \mathbf{z} \in \mathcal{Y}} \varphi_K(\mathbf{x}, \mathbf{z}). \tag{6}$$

This section is mainly devoted to the convergence of solutions of approximation problems in Eq. (6) towards those of the original BLO in Eq. (5).

**Assumption 3.1** *We make following standing assumptions throughout this section.*

 (1) *$F, f : \mathbb{R}^n \times \mathbb{R}^m \to \mathbb{R}$ are continuous functions.*

 (2) *$\nabla f$ is continuous and $\nabla_{\mathbf{y}} f$ is $L_f$ Lipschitz continuous with respect to $\mathbf{y}$ for any $\mathbf{x} \in \mathcal{X}$.*

 (3) *$\mathcal{X}$ and $\mathcal{Y}$ are convex compact sets.*

 (4) *$\mathcal{S}(\mathbf{x})$ is nonempty for any $\mathbf{x} \in \mathcal{X}$.*

 (5) *For any $(\bar{\mathbf{x}}, \bar{\mathbf{y}})$ minimizing $F(\mathbf{x}, \mathbf{y})$ over constraints $\mathbf{x} \in \mathcal{X}, \mathbf{y} \in \mathcal{Y}$ and $\mathbf{y} \in \hat{\mathcal{S}}(\mathbf{x})$, it holds that $\bar{\mathbf{y}} \in \mathcal{S}(\mathbf{x})$.*

Note that $\hat{\mathcal{S}}(\mathbf{x})$ denotes the set of LL stationary points, i.e., $\hat{\mathcal{S}}(\mathbf{x}) = \{\mathbf{y} \in \mathcal{Y} | 0 = \nabla_{\mathbf{y}} f(\mathbf{x}, \mathbf{y}) + \mathcal{N}_{\mathcal{Y}}(\mathbf{y})\}$. It should be noticed that $\mathbf{y} \in \hat{\mathcal{S}}(\mathbf{x})$ if and only if $\mathcal{R}_{\alpha}(\mathbf{x}, \mathbf{y}) = 0$. Assumption 3.1 is standard in bi-level optimziation related literature, which will be shown to be satisfied for the numerical example given in Section 4.1.

As discussed in the preceding section, for each $\mathbf{x} \in \mathcal{X}, \mathbf{z} \in \mathcal{Y}$ and $K > 0$, there exists at least a $\tilde{K}$ such that along the selection $\mathbf{y}_{\tilde{K}}(\mathbf{x}, \mathbf{z})$, $\mathcal{R}_{\alpha}(\mathbf{x}, \mathbf{y}_{\tilde{K}}(\mathbf{x}, \mathbf{z}))$ uniformly converges to zero w.r.t. $\mathbf{x}$ and $\mathbf{z}$, as $K$ tending infinity. We next specifically show the existence of such index $\tilde{K}$. To this end, $\tilde{K}$ can be chosen by optimizing $\|\mathcal{R}_{\alpha}(\mathbf{x}, \mathbf{y}_k(\mathbf{x}, \mathbf{z}))\|$ among the indices $k = 0, 1, ..., K$. In particular, as stated in the following lemma, $\|\mathcal{R}_{\alpha}(\mathbf{x}, \mathbf{y}_{\tilde{K}}(\mathbf{x}, \mathbf{z}))\|$ uniformly decreases with a $\frac{1}{\sqrt{K}}$ rate on $\mathcal{X} \times \mathcal{Y}$ as $K$ increases.

**Lemma 3.1** *Let $\{\mathbf{y}_k(\mathbf{x}, \mathbf{z})\}$ be the sequence defined in Eq. (4) with $\alpha_{\mathbf{y}}^k \in [\underline{\alpha}_{\mathbf{y}}, \overline{\alpha}_{\mathbf{y}}] \subset (0, \frac{2}{L_f})$, there exists $C_f > 0$ such that*

$$\min_{0 \leq k \leq K} \|\mathcal{R}_{\underline{\alpha}_{\mathbf{y}}}(\mathbf{x}, \mathbf{y}_k(\mathbf{x}, \mathbf{z}))\| \leq \frac{C_f}{\sqrt{K+1}}, \quad \forall \mathbf{x} \in \mathcal{X}, \mathbf{z} \in \mathcal{Y}. \tag{7}$$

As shown in the Appendix, the proof of Lemma 3.1 for the existence cannot offer us an explicit identification of the exact selection of $\tilde{K}$. Alternatively, we construct the approximation by a pessimistic trajectory truncation strategy, minimizing the worst case of all selections of $\{\mathbf{y}_k(\mathbf{x}, \mathbf{z})\}$, i.e., $\varphi_K(\mathbf{x}, \mathbf{z})$. By further solving the approximated problems $\min \varphi_K(\mathbf{x}, \mathbf{z})$, we shall provide a lower bound estimation for the optimal value of the BLO problem in Eq. (5).

**Lemma 3.2** *Let $(\mathbf{x}_K, \mathbf{z}_K) \in \underset{\mathbf{x} \in \mathcal{X}, \mathbf{z} \in \mathcal{Y}}{\operatorname{argmin}} \varphi_K(\mathbf{x}, \mathbf{z})$, then*

$$\varphi_K(\mathbf{x}_K, \mathbf{z}_K) \leq \inf_{\mathbf{y} \in \hat{\mathcal{S}}(\mathbf{x})} F(\mathbf{x}, \mathbf{y}), \quad \forall \mathbf{x} \in \mathcal{X}. \tag{8}$$

Upon together with the uniform convergence result in Lemma 3.1, the gap between the lower bound provided by $\min \varphi_K(\mathbf{x}, \mathbf{z})$ and the true optimal value of the BLO problem in Eq. (5) eventually vanishes. To fill in this gap and present the main convergence result of our proposed IAPTT-GM,

we need the continuity of $\mathcal{R}_\alpha(\mathbf{x}, \mathbf{y})$. Indeed, it follows from [43, Theorem 6.42] that $\mathtt{Proj}_{\mathcal{Y}}$ is continuous. Combined with the assumed continuity of $\nabla_{\mathbf{y}} f(\mathbf{x}, \mathbf{y})$, we get the desired continuity of $\mathcal{R}_\alpha(\mathbf{x}, \mathbf{y})$ immediately.

**Theorem 3.1** *Let $\{\mathbf{y}_k(\mathbf{x}, \mathbf{z})\}$ be the sequence generated by Eq. (4) with $\alpha_{\mathbf{y}}^k \in [\underline{\alpha}_{\mathbf{y}}, \overline{\alpha}_{\mathbf{y}}] \subset (0, \frac{2}{L_f})$, and $(\mathbf{x}_K, \mathbf{z}_K) \in \underset{\mathbf{x} \in \mathcal{X}, \mathbf{z} \in \mathcal{Y}}{\operatorname{argmin}} \varphi_K(\mathbf{x}, \mathbf{z})$, then we have:*

*(1) Any limit point $\bar{\mathbf{x}}$ of the sequence $\{\mathbf{x}_K\}$ is the solution to BLO in Eq. (1), that is $\bar{\mathbf{x}} \in \underset{\mathbf{x} \in \mathcal{X}}{\operatorname{argmin}} \varphi(\mathbf{x})$.*

*(2) $\underset{\mathbf{x} \in \mathcal{X}, \mathbf{z} \in \mathcal{Y}}{\inf} \varphi_K(\mathbf{x}, \mathbf{z}) \to \underset{\mathbf{x} \in \mathcal{X}}{\inf} \varphi(\mathbf{x})$ as $K \to \infty$.*

In the above theorem, we justify the global solutions convergence of the approximated problems. We next derive a convergence characterization regarding the local minimums of the approximated problems. In particular, the next theorem shows that any limit point of the local minimums of approximated problems is in some sense a local minimum of the bilevel problem in Eq. (1).

**Theorem 3.2** *Let $\{\mathbf{y}_k(\mathbf{x}, \mathbf{z})\}$ be the sequence generated by Eq. (4) with $\alpha_{\mathbf{y}}^k \in [\underline{\alpha}_{\mathbf{y}}, \overline{\alpha}_{\mathbf{y}}] \subset (0, \frac{2}{L_f})$, and $(\mathbf{x}_K, \mathbf{z}_K)$ be a local minimum of $\varphi_K(\mathbf{x}, \mathbf{z})$ with uniform neighborhood modulus $\delta > 0$, i.e.,*

$$\varphi_K(\mathbf{x}_K, \mathbf{z}_K) \le \varphi_K(\mathbf{x}, \mathbf{z}), \quad \forall (\mathbf{x}, \mathbf{z}) \in \mathbb{B}_\delta(\mathbf{x}_K, \mathbf{z}_K) \cap \mathcal{X} \times \mathcal{Y}.$$

*Then we have that for any limit point $(\bar{\mathbf{x}}, \bar{\mathbf{z}})$ of the sequence $\{(\mathbf{x}_K, \mathbf{z}_K)\}$, there exists a limit point $\bar{\mathbf{y}}$ of the sequence $\{\mathbf{y}_K(\mathbf{x}_K, \mathbf{z}_K)\}$ such that $\bar{\mathbf{y}} \in \hat{\mathcal{S}}(\bar{\mathbf{x}})$ and $(\bar{\mathbf{x}}, \bar{\mathbf{y}})$ satisfies that there exists $\tilde{\delta} > 0$ such that*

$$F(\bar{\mathbf{x}}, \bar{\mathbf{y}}) \le F(\mathbf{x}, \mathbf{z}), \quad \forall (\mathbf{x}, \mathbf{z}) \in \mathbb{B}_{\tilde{\delta}}(\bar{\mathbf{x}}, \bar{\mathbf{z}}) \cap \{\mathbf{x} \in \mathcal{X}, \mathbf{z} \in \mathcal{Y} \mid \mathbf{z} \in \hat{\mathcal{S}}(\mathbf{x})\}.$$

### 3.2 Theoretical Findings of IA-GM (A) for BLO with LLC

A byproduct of our study, which has its own interest, is that thanks to the involved initialization auxiliary, our theory can improve those existing results for classical gradient methods with accelerated gradient descent dynamical iterations under LLC. Nesterov's acceleration technique [44] has been used widely for solving convex optimization problem and it greatly improves the convergence rate of gradient descent. To illustrate our result, we will take the Nesterov's acceleration proximal gradient method [45] as the embedded optimization dynamics in classical gradient method for example.

Thanks to the LLC setting, $\mathcal{R}_\alpha(\mathbf{x}, \mathbf{y}_K(\mathbf{x}, \mathbf{z}))$ may uniformly converge to zero w.r.t. UL variable $x$ and auxiliary variable $\mathbf{z}$, thus the pessimistic trajectory truncation operation can be removed. Subsequently, we slightly simplify our algorithm that $\bar{k}$ is simply taken as $K$, thus the approximation objective admits a succinct form, i.e., $F(\mathbf{x}, \mathbf{y}_K(\mathbf{x}, \mathbf{z}))$.

In summary, by constructing $\mathbf{y}_k(\mathbf{x}, \mathbf{z})$ through following Nesterov's acceleration dynamics,

$$\mathbf{y}_0(\mathbf{x}, \mathbf{z}) = \mathbf{z}, \qquad t_0 = 1, \quad t_{k+1} = \frac{1 + \sqrt{1 + t_k^2}}{2}, \ k = 0, \cdots, K-1,$$

$$\mathbf{u}^{k+1}(\mathbf{x}, \mathbf{z}) = \mathbf{y}^{k+1}(\mathbf{x}, \mathbf{z}) + \left( \frac{t_k - 1}{t_{k+1}} \right) (\mathbf{y}^{k+1}(\mathbf{x}, \mathbf{z}) - \mathbf{y}^k(\mathbf{x}, \mathbf{z})), \ k = 0, \cdots, K-1, \qquad (9)$$

$$\mathbf{y}_{k+1}(\mathbf{x}, \mathbf{z}) = \mathtt{Proj}_{\mathcal{Y}} \left( \mathbf{u}^k(\mathbf{x}, \mathbf{z}) - \alpha \nabla_{\mathbf{y}} f(\mathbf{x}, \mathbf{u}^k(\mathbf{x}, \mathbf{z})) \right), \ k = 0, \cdots, K-1,$$

where $\alpha > 0$ is the step size, we propose an accelerated Gradient-based Method with Initialization Auxiliary, named IA-GM(A), via minimizing the approximation objective function,

$$\min_{\mathbf{x} \in \mathcal{X}, \mathbf{z} \in \mathcal{Y}} \phi_K(\mathbf{x}, \mathbf{z}) := F(\mathbf{x}, \mathbf{y}_K(\mathbf{x}, \mathbf{z})).$$

The detailed description of the proposed IA-GM(A) is stated in the Supplemental Material.

We suppose Assumption 3.1(1)-(4) are satisfied throughout this subsection. The convergence result of our proposed IA-GM(A) with LLC assumption is given as below.

**Theorem 3.3** *Assume that the generated sequence $\{\mathbf{y}_k(\mathbf{x}, \mathbf{z})\}$ satisfies that $\mathbf{y}_k(\mathbf{x}, \mathbf{z}) \in \mathcal{Y}$, and $\mathbf{y}_k(\mathbf{x}, \mathbf{z}) = \mathbf{z}$ for any $\mathbf{z} \in \mathcal{S}(\mathbf{x})$, $\mathbf{x} \in \mathcal{X}$, and either*

(a) *for any $\epsilon > 0$, there exists $k(\epsilon) > 0$ such that whenever $K > k(\epsilon)$,*

$$\sup_{\mathbf{x}\in\mathcal{X},\mathbf{z}\in\mathcal{Y}} \{f(\mathbf{x},\mathbf{y}_K(\mathbf{x},\mathbf{z})) - f^*(\mathbf{x})\} \leq \epsilon$$

   *whenever $K > k(\epsilon)$, or*

(b) *there exists $\alpha > 0$, for any $\epsilon > 0$, there exists $k(\epsilon) > 0$ such that,*

$$\sup_{\mathbf{x}\in\mathcal{X},\mathbf{z}\in\mathcal{Y}} \|\mathcal{R}_\alpha(\mathbf{x},\mathbf{y}_K(\mathbf{x},\mathbf{z}))\| \leq \epsilon.$$

*Let $(\mathbf{x}_K,\mathbf{z}_K) \in \underset{\mathbf{x}\in\mathcal{X},\mathbf{z}\in\mathcal{Y}}{\operatorname{argmin}} \phi_K(\mathbf{x},\mathbf{z}) := F(\mathbf{x},\mathbf{y}_K(\mathbf{x},\mathbf{z}))$, then we have*

(1) *any limit point $\bar{x}$ of the sequence $\{\mathbf{x}_K\}$ satisfies that $\bar{\mathbf{x}} \in \underset{\mathbf{x}\in\mathcal{X}}{\operatorname{argmin}}\varphi(\mathbf{x})$, i.e., $\bar{x}$ is the solution to BLO* (1).

(2) $\underset{\mathbf{x}\in\mathcal{X},\mathbf{z}\in\mathcal{Y}}{\inf} \phi_K(\mathbf{x},\mathbf{z}) \to \underset{\mathbf{x}\in\mathcal{X}}{\inf} \varphi(\mathbf{x})$ *as $K \to \infty$.*

Next, we show that the Nesterov's acceleration dynamics satisfy all the assumptions required in the above convergence theorem.

**Theorem 3.4** *Let $\{\mathbf{y}_k(\mathbf{x},\mathbf{z})\}$ be the sequence generated by Nesterov's acceleration dynamics in Eq. (9) with $\alpha = \frac{1}{L_f}$. Then $\{\mathbf{y}_k(\mathbf{x},\mathbf{z})\}$ satisfies all the assumptions required by Theorem 3.3.*

**Remark 3.1** *Our convergence result Theorem 3.3 is not only for $\mathbf{y}_k$ generated by Nesterov's acceleration dynamics. It is a general convergence result that is applicable for the case where $\mathbf{y}_k$ is generated by other dynamics. And the assumptions required in Theorem 3.3 is weak enough to be satisfied by the dynamics introduced by many first-order methods on convex LL problem in Eq. 2.*

## 4 Experimental Results

In this section, we first verify the theoretical convergence results on non-convex numerical problems compared with existing EG methods and IG methods. Then we test the performance of IAPTT-GM and demonstrate its generalizability to real-world BLO problems with non-convex followers, which are caused by non-convex regularization and neural network structures. In addition, we further validate the performance of the accelerated version (i.e., IA-GM (A)) under LLC with numerical examples and data hyper-cleaning tasks [2].

### 4.1 Numerical Verification

To verify the convergence property under assumptions provided in Section 3, we consider the following non-convex BLO problem:

$$\min_{x\in\mathcal{X},y\in\mathbb{R}} x + xy, \quad s.t. \quad y \in \underset{y\in\mathcal{Y}}{\operatorname{argmin}} - \sin(xy), \tag{10}$$

where $\mathcal{X} = [1,10]$ and $\mathcal{Y} = [-2,2]$. Given any $x \in \mathcal{X}$, it satisfies $\operatorname{argmin}_{y\in\mathcal{Y}} - \sin(xy) = \{(2k\pi + \pi/2)/x \mid k \in \mathbb{Z}\} \cap \mathcal{Y}$ and $\min_{y\in\mathcal{Y}} - \sin(xy) = -1$. The unique solution is $(x^*,y^*) = (11\pi/4, -2)$. It should be noted that the LL problem of Eq. (10) has multiple global minima, which can significantly show the advantage of initialization auxiliary technique. It can be easily verified that the above toy example satisfies Assumption 3.1.

In Figure 1, we separately compared IAPTT-GM with EG methods such as RHG [3], BDA [23], IG methods such as LS [28], NS [7] and IA-GM. From Figure 1.(a) to Figure 1.(b), we can observe that different initialization points only slightly affect the convergence speed of IAPTT-GM. With initialization points distant from $(x^*,y^*)$, IAPTT-GM can still achieve optimal solution of UL variables and optimal objective value, while other methods fail to converge to the true solution. In Figure 1.(g) and Figure 1.( h), we compare the performances of IAPTT-GM with IA-GM, which has no convergence guarantee without LLC assumption. As shown, IA-GM fails to converge to the true solution eventually, which validates the necessity of PTT technique and the effectiveness of IAPTT-GM.

---

[2]The code is available at http://github.com/vis-opt-group/IAPTT-GM.

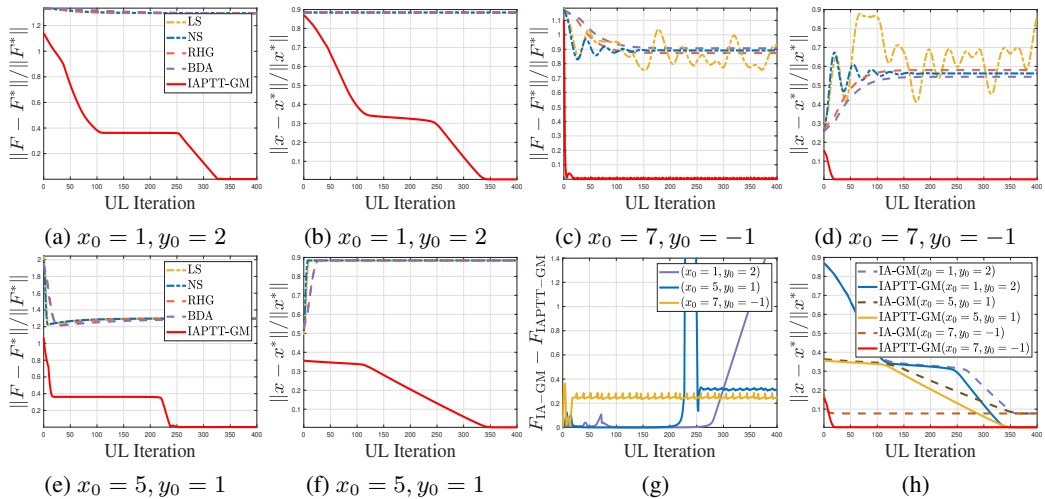

(a) $x_0 = 1, y_0 = 2$    (b) $x_0 = 1, y_0 = 2$    (c) $x_0 = 7, y_0 = -1$    (d) $x_0 = 7, y_0 = -1$

(e) $x_0 = 5, y_0 = 1$    (f) $x_0 = 5, y_0 = 1$    (g)    (h)

Figure 1: Illustrating the convergence behavior of $\|F - F^*\|/\|F^*\|$ and $\|x - x^*\|/\|x^*\|$ as the training proceeds. $F_{\text{IAPTT-GM}}$ and $F_{\text{IA-GM}}$ denote the UL objectives of IAPTT-GM and IA-GM, respectively. Three representative initialization points for UL and LL variables are $(x_0, y_0) = (1, 2)$, $(x_0, y_0) = (5, 1)$, $(x_0, y_0) = (7, -1)$.

**Runtime and Memory Analysis.** In Figure 2, we report the average steps $\bar{k}$ of IAPTT-GM for PTT and the iterative speed as the UL iteration increases. In comparison with IA-GM, which uses default $K$ for the LL optimization loop, the changing $\bar{k}$ for IAPTT-GM leads to less iterations for the backward recurrent propagation and thus faster iterative updates during optimization. Although IA introduces additional variables and iterations, the PTT technique can choose a small $\bar{k}$, thus shortens the back-propagation trajectory for computing the UL gradient (see Figure 2). As can be seen in Table 1, the memory required by our IAPTT-GM is less than NS, LS, and BDA and the same as that for RHG. As for the runtime, IAPTT-GM is a bit slower than RHG and NS, and faster than BDA. But please notice that the performance and the theoretical properties of IAPTT-GM are better than these existing approaches.

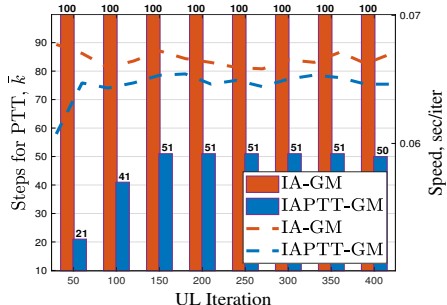

Figure 2: Illustrating average steps for PTT technique and average running speed of the numerical example. Note that we conduct the experiments using more LL iterations so as to reduce the measurement error.

Table 1: Memory and runtime of existing methods for solving the above BLO problem. We conduct the experiments using the same parameter settings in Section C of the supplementary materials.

| Metrics | LS | NS | RHG | BDA | IAPTT-GM |
|---|---|---|---|---|---|
| Memory (GB) | 10.426 | 10.387 | 10.153 | 10.154 | 10.153 |
| Runtime (Sec) | 5.120 | 10.815 | 9.990 | 16.800 | 10.835 |

## 4.2 BLO with Non-convex Followers in Different Application Scenarios

To cover various real-world BLO application scenarios, we consider two categories of non-convex LL problems caused by non-convex regularization term and neural network architectures, which refer to few-shot classification and data hyper-cleaning tasks, respectively. Please note that the set constraint $\mathcal{Y}$ is only used to guarantee the completeness of our theoretical analysis in Section 3. In application scenarios, we can just consider the constraint as an extra large set, so that all the variables are automatically in this feasible set. In this way, it is natural to ignore the projection operation in practical computations.

#### 4.2.1 Few-Shot Classification: Non-convex LL Objective

In few-shot learning, to be more specific, N-way M-shot classification tasks [46], provided with M samples from each class, we train models to take advantage of prior data from similar tasks to quickly classify unseen instances from these N classes. Following the experimental protocol [47], the model parameters are separated into two parts: the hyper representation module (parameterized by $\mathbf{x}$) shared by all the tasks and the last classifier (parameterized by $\mathbf{y}^j$) for $j$-th task. Define the meta training dataset as $\mathcal{D} = \{\mathcal{D}^j\}$, where $\mathcal{D}^j = \mathcal{D}_{\mathtt{tr}}^j \bigcup \mathcal{D}_{\mathtt{val}}^j$ corresponds to the $j$-th task.

The cross-entropy loss function is widely used for UL and LL objectives. Referring to Section 3, IAPTT-GM covers the convergence results with non-convex LL model, thus allowing flexible design of the LL objective function. For instance, while non-convex regularization terms, e.g., $\ell_q$ regularization with $0 < q < 1$, have shown effectiveness to help the LL model converge and avoid over-fitting, existing methods can only guarantee the convergence when $q \geq 1$, thus almost provide no support for non-convex objectives. We consider the LL subproblem with non-convex loss functions by adding $\ell_q$ regularization [3]. Then the UL and LL subproblems can be written as

$$F\left(\mathbf{x}, \{\mathbf{y}^j\}\right) = \sum_j \ell\left(\mathbf{x}, \mathbf{y}^j; \mathcal{D}_{\mathtt{val}}^j\right), \quad f\left(\mathbf{x}, \{\mathbf{y}^j\}\right) = \sum_j \ell\left(\mathbf{x}, \mathbf{y}^j; \mathcal{D}_{\mathtt{tr}}^j\right) + \|\mathbf{y}^j\|_q. \quad (11)$$

Table 2: Mean test accuracy of 5-way classification on tieredImageNet and miniImageNet, and the $\pm$ represents 95% confidence intervals over tasks.

| Methods | Backbone | MiniImagenet | | TieredImagenet | |
|---|---|---|---|---|---|
| | | 5-way 1-shot | 5-way 5-shot | 5-way 1-shot | 5-way 5-shot |
| Proto Net | ConvNet-4 | $49.42 \pm 1.84$ | $68.20 \pm 0.66$ | $53.31 \pm 0.89$ | $72.69 \pm 0.74$ |
| Relation Net | ConvNet-4 | $50.44 \pm 0.82$ | $65.32 \pm 0.70$ | $54.48 \pm 0.93$ | $65.32 \pm 0.70$ |
| MAML | ConvNet-4 | $48.70 \pm 0.75$ | $63.11 \pm 0.11$ | $49.06 \pm 0.50$ | $67.48 \pm 0.47$ |
| RHG | ConvNet-4 | $48.89 \pm 0.81$ | $63.02 \pm 0.70$ | $49.63 \pm 0.67$ | $66.14 \pm 0.57$ |
| T-RHG | ConvNet-4 | $47.67 \pm 0.82$ | $63.70 \pm 0.76$ | $50.79 \pm 0.69$ | $67.39 \pm 0.60$ |
| BDA | ConvNet-4 | $49.08 \pm 0.82$ | $62.17 \pm 0.70$ | $51.56 \pm 0.68$ | $68.21 \pm 0.58$ |
| MAML | ResNet-12 | $51.03 \pm 0.50$ | $68.26 \pm 0.47$ | $58.58 \pm 0.49$ | $71.24 \pm 0.43$ |
| RHG | ResNet-12 | $50.54 \pm 0.85$ | $64.53 \pm 0.68$ | $58.19 \pm 0.76$ | $75.20 \pm 0.60$ |
| IAPTT-GM | ResNet-12 | $\mathbf{56.69 \pm 0.66}$ | $\mathbf{70.21 \pm 0.55}$ | $\mathbf{60.71 \pm 0.77}$ | $\mathbf{75.85 \pm 0.59}$ |

Detailed information about the datasets and network architectures can be found in the supplementary materials. We report results of IAPTT-GM and various mainstream methods, e.g., Prototypical Network [48], Relation Net [49] and T-RHG [16] on miniImageNet [47] and tieredImageNet [50] datasets with two different backbones [6, 51] in Table 2. As it is shown, our proposed method outperforms state-of-the-art methods on both 5-way 1-shot and 5-way 5-shot tasks.

#### 4.2.2 Data Hyper-Cleaning: Non-convex LL Architecture Structure

Data hyper-cleaning [3] aims to cleanup the corrupted data with noise label. According to [16], the dataset is randomly split to three disjoint subsets: $\mathcal{D}_{\mathtt{tr}}$ for training, $\mathcal{D}_{\mathtt{val}}$ for validation and $\mathcal{D}_{\mathtt{test}}$ for testing, then a fixed proportion of the training samples in $\mathcal{D}_{\mathtt{tr}}$ is randomly corrupted.

Following the classical experimental protocol [3], we choose cross-entropy as the loss function $\ell$, and the UL and LL subproblem take the form of

$$F(\mathbf{x}, \mathbf{y}) = \sum_{(\mathbf{u}_i, \mathbf{v}_i) \in \mathcal{D}_{\mathtt{val}}} \ell\left(\mathbf{y}(\mathbf{x}); \mathbf{u}_i, \mathbf{v}_i\right), \quad f(\mathbf{x}, \mathbf{y}) = \sum_{(\mathbf{u}_i, \mathbf{v}_i) \in \mathcal{D}_{\mathtt{tr}}} [\sigma(\mathbf{x})]_i \ell\left(\mathbf{y}; \mathbf{u}_i, \mathbf{v}_i\right), \quad (12)$$

where $(\mathbf{u}_i, \mathbf{v}_i)$ denotes the data pair and $\sigma(\mathbf{x})$ represents the element-wise sigmoid function on $\mathbf{x}$. We define the hyperparameter $\mathbf{x}$ as a vector being trained to label the noisy data, of which the dimension equals to the number of training samples. The LL variables parameterized by $\mathbf{y}$ contain the weights and bias of fully connected layers.

---

[3]Here, we use the smoothing regularization term defined as $\|\boldsymbol{w}\|_q = (\|\boldsymbol{w}\|_2 + \|\boldsymbol{\varepsilon}\|_2)^{q/2}$, where $0 < q < 1$.

Note that existing methods consider convex a single fully connected layer as the LL model, while more complex neural network structure is not applicable. Under our assumption without LLC, we employ two fully connected layers as the LL network architecture.

In Table 3, we compare IAPTT-GM with IG methods (e.g., LS, NS) and EG methods (e.g., RHG, T-RHG [16]). As it is shown, IAPTT-GM achieves better test performance of both accuracy and F1 score on two datasets, including MNIST [46] and FashionMNIST [52]. Our theoretical results also show that the performance improvement comes from PTT and IA techniques to overcome non-convex LL subproblems.

Table 3: Reporting results of existing methods for solving data hyper-cleaning tasks. Acc. and F1 score denote the test accuracy and the harmonic mean of the precision and recall, respectively.

| Method | MNIST | | FashionMNIST | |
|---|---|---|---|---|
| | Acc. | F1 score | Acc. | F1 score |
| LS | 89.19 | 85.96 | 83.15 | 85.13 |
| NS | 87.54 | 89.58 | 81.37 | 87.28 |
| RHG | 87.90 | 89.36 | 81.91 | 87.12 |
| T-RHG | 88.57 | 89.77 | 81.85 | 86.76 |
| BDA | 87.15 | 90.38 | 79.97 | 88.24 |
| IAPTT-GM | **90.88** | **91.57** | **83.67** | **90.37** |

### 4.3 Evaluations of IA-GM (A) for BLOs under LLC

In addition to non-convex BLO problems, we also raise concerns about the acceleration strategy of our method with LLC condition. We first consider the following BLO with LLC condition [23]:

$$\min_{\mathbf{x} \in \mathcal{X}} \|\mathbf{x} - \mathbf{y}_2\|^4 + \|\mathbf{y}_1 - \mathbf{e}\|^4, \quad s.t. \quad (\mathbf{y}_1, \mathbf{y}_2) \in \arg\min_{\mathbf{y}_1 \in \mathbb{R}^n, \mathbf{y}_2 \in \mathbb{R}^n} \frac{1}{2} \|\mathbf{y}_1\|^2 - \mathbf{x}^\top \mathbf{y}_1, \quad (13)$$

where $n = 50$, $\mathcal{X} = [-100, 100] \times \cdots [-100, 100] \subset \mathbb{R}^n$, and $\mathbf{e}$ represents the vector whose elements are all equal to 1. The optimal solution for this problem is $\mathbf{x}^* = \mathbf{e}, \mathbf{y}_1^* = \mathbf{e}, \mathbf{y}_2^* = \mathbf{e}$.

As shown in Section 3.2, our method IA-GM (A) incorporates Nesterov's acceleration strategy for solving Eq. (2). Note that with LLC assumption, IA-GM maintains the convergence property.

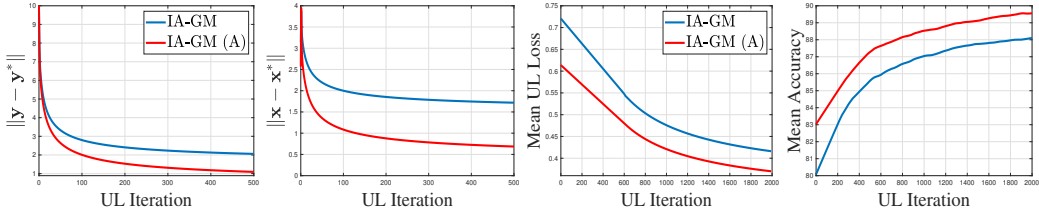

(a) Convex toy regression  (b) Convex toy regression  (c) Data hyper-cleaning  (d) Data hyper-cleaning

Figure 3: The left two subfigures report the curves of $\|\mathbf{y} - \mathbf{y}^*\|$ and $\|\mathbf{x} - \mathbf{x}^*\|$ for IA-GM and IA-GM (A). The figures on the right illustrate the results of mean UL loss and accuracy on the convex data hyper-cleaning problems.

As illustrated in the first two figures in Figure 3, IA-GM (A) shows significant improvement of convergence speed on UL and LL variables, which verifies the convergence results of Theorem 3.4 under LLC. We further study the convex data hyper-cleaning problem, which simply implements single fully connected layer as the network structure. From the right half of Figure 3, we can easily find that IA-GM (A) also performs better than IA-GM on real-world applications.

## 5 Conclusion

This paper presents a generic first-order algorithmic framework named IAPTT-GM to solve BLO problems with non-convex follower. We introduce two features, initialization auxiliary and pessimistic trajectory truncation operation to guarantee the convergence without the LLC hypothesis and achieves better performance on various applications. Meanwhile, we also validate the performance and speed improvement for BLO with LLC condition.

## Acknowledgments and Disclosure of Funding

This work is partially supported by the National Natural Science Foundation of China (Nos. 61922019, 11971220), the National Key R&D Program of China (2020YFB1313503), LiaoNing Revitalization Talents Program (XLYC1807088), the Shenzhen Science and Technology Program (No. RCYX20200714114700072), the Fundamental Research Funds for the Central Universities, the Pacific Institute for the Mathematical Sciences (PIMS), the Stable Support Plan Program of Shenzhen Natural Science Fund (No. 20200925152128002) and the Guangdong Basic and Applied Basic Research Foundation 2019A1515011152.

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
