The supplemental material is organized as follow. Detailed proofs of theoretical results in Section 3.1 and Section 3.2 are provided in Section A and Section B, respectively. Section C presents configurations of computing devices and detailed settings (e.g., data splits, hyper-parameters) of numerical experiments given in Section 4 of the main paper.

## A    Proof of results in Section 3.1

### A.1    Proof of Lemma 3.1

*Proof.* First, as $\mathcal{X}$ and $\mathcal{Y}$ are compact sets and $f$ is continuous on $\mathcal{X} \times \mathcal{Y}$, there exist constants $m$, $M$ such that $m \leq f(\mathbf{x}, \mathbf{y}) \leq M$ for any $(\mathbf{x}, \mathbf{y}) \in \mathcal{X} \times \mathcal{Y}$. According to [1, Lemma 10.4], we have

$$\left( \frac{1}{\alpha_{\mathbf{y}}^k} - \frac{L_f}{2} \right) \|\mathcal{R}_{\alpha_{\mathbf{y}}^k}(\mathbf{x}, \mathbf{y}_k(\mathbf{x}, \mathbf{z}))\|^2 \leq f(\mathbf{x}, \mathbf{y}_k(\mathbf{x}, \mathbf{z})) - f(\mathbf{x}, \mathbf{y}^{k+1}(\mathbf{x}, \mathbf{z})), \;\; \forall k \geq 0.$$

Since $\alpha_{\mathbf{y}}^k \in [\underline{\alpha}_{\mathbf{y}}, \overline{\alpha}_{\mathbf{y}}] \subset (0, \frac{2}{L_f})$, it follows from [1, Theorem 10.9] that $\|\mathcal{R}_{\underline{\alpha}_{\mathbf{y}}}(\mathbf{x}, \mathbf{y}_k(\mathbf{x}, \mathbf{z}))\| \leq \|\mathcal{R}_{\alpha_{\mathbf{y}}^k}(\mathbf{x}, \mathbf{y}_k(\mathbf{x}, \mathbf{z}))\|$, and thus

$$\|\mathcal{R}_{\underline{\alpha}_{\mathbf{y}}}(\mathbf{x}, \mathbf{y}_k(\mathbf{x}, \mathbf{z}))\|^2 \leq \frac{1}{(1/\overline{\alpha}_{\mathbf{y}} - L_f/2)} \left( f(\mathbf{x}, \mathbf{y}_k(\mathbf{x}, \mathbf{z})) - f(\mathbf{x}, \mathbf{y}^{k+1}(\mathbf{x}, \mathbf{z})) \right), \;\; \forall k \geq 0.$$

Summing the above inequality from $k = 0$ to $K$, we have

$$\sum_{k=0}^{K} \|\mathcal{R}_{\underline{\alpha}_{\mathbf{y}}}(\mathbf{x}, \mathbf{y}_k(\mathbf{x}, \mathbf{z}))\|^2 \leq \frac{1}{(1/\overline{\alpha}_{\mathbf{y}} - L_f/2)} \left( f(\mathbf{x}, \mathbf{y}^0(\mathbf{x}, \mathbf{z})) - f(\mathbf{x}, \mathbf{y}^{K+1}(\mathbf{x}, \mathbf{z})) \right).$$

Since $\mathbf{y}^k(\mathbf{x}, \mathbf{z}) \in \mathcal{Y}$ for any $k$, $m \leq f(\mathbf{x}, \mathbf{y}^k(\mathbf{x}, \mathbf{z})) \leq M$ for any $\mathbf{x} \in \mathcal{X}, \mathbf{z} \in \mathcal{Y}$ and $k \geq 0$. Then we can obtain from the above inequality that

$$\min_{0 \leq k \leq K} \|\mathcal{R}_{\underline{\alpha}_{\mathbf{y}}}(\mathbf{x}, \mathbf{y}_k(\mathbf{x}, \mathbf{z}))\| \leq \sqrt{\frac{M - m}{(1/\overline{\alpha}_{\mathbf{y}} - L_f/2)(K + 1)}}, \quad \forall \mathbf{x} \in \mathcal{X}, \mathbf{z} \in \mathcal{Y}.$$

The conclusion follows by letting $C_f = \sqrt{\frac{M-m}{1/\overline{\alpha}_{\mathbf{y}} - L_f/2}}$. $\qquad\square$

### A.2    Proof of Lemma 3.2

*Proof.* For any $\mathbf{x} \in \mathcal{X}$, and any $\epsilon > 0$, there exists $\mathbf{y}_\epsilon \in \hat{\mathcal{S}}(\mathbf{x})$ such that $F(\mathbf{x}, \mathbf{y}_\epsilon) \leq \inf_{\mathbf{y} \in \hat{\mathcal{S}}(\mathbf{x})} F(\mathbf{x}, \mathbf{y}) + \epsilon$. As $\mathbf{y}_\epsilon \in \hat{\mathcal{S}}(x)$, then $\mathcal{R}_\alpha(\mathbf{x}, \mathbf{y}_\epsilon) = 0$ for any $\alpha > 0$ and thus $\mathbf{y}_k(\mathbf{x}, \mathbf{y}_\epsilon) = \mathbf{y}_\epsilon$ for any $k \geq 0$. Since $\mathbf{y}_\epsilon \in \mathcal{Y}$, we have

$$\varphi_K(\mathbf{x}_K, \mathbf{z}_K) \leq \varphi_K(\mathbf{x}, \mathbf{y}_\epsilon) = \max_{1 \leq k \leq K} \{F(\mathbf{x}, \mathbf{y}_k(\mathbf{x}, \mathbf{y}_\epsilon))\} = F(\mathbf{x}, \mathbf{y}_\epsilon) \leq \inf_{\mathbf{y} \in \hat{\mathcal{S}}(\mathbf{x})} F(\mathbf{x}, \mathbf{y}) + \epsilon.$$

The conclusion follows by letting $\epsilon \to 0$ in above inequality. $\qquad\square$

### A.3    Proof of Theorem 3.1

*Proof.* For any $K > 0$, we define $i(K) := \operatorname{argmin}_{0 \leq k \leq K} \|\mathcal{R}_{\underline{\alpha}_{\mathbf{y}}}(\mathbf{x}, \mathbf{y}_k(\mathbf{x}, \mathbf{z}))\|$. For any limit point $\bar{\mathbf{x}}$ of the sequence $\{\mathbf{x}_K\}$, let $\{\mathbf{x}_l\}$ be a subsequence of $\{\mathbf{x}_K\}$ such that $\mathbf{x}_l \to \bar{\mathbf{x}} \in \mathcal{X}$. As $\{\mathbf{y}_{i(K)}(\mathbf{x}_K, \mathbf{z}_K)\} \subset \mathcal{Y}$ and $\mathcal{Y}$ is compact, we can find a subsequence $\{\mathbf{x}_j\}$ of $\{\mathbf{x}_l\}$ satisfying $\mathbf{y}_{i(j)}(\mathbf{x}_j, \mathbf{z}_j) \to \bar{\mathbf{y}}$ for some $\bar{\mathbf{y}} \in \mathcal{Y}$. It follows from Lemma 3.1 that for any $\epsilon > 0$, there exists $J(\epsilon) > 0$ such that for any $j > J(\epsilon)$, we have

$$\|\mathcal{R}_{\underline{\alpha}_{\mathbf{y}}}(\mathbf{x}_j, \mathbf{y}_{i(j)}(\mathbf{x}_j, \mathbf{z}_j))\| \leq \epsilon.$$

By letting $j \to \infty$, and since $\mathcal{R}_\alpha(\mathbf{x}, \mathbf{y})$ is continuous, we have

$$\|\mathcal{R}_{\underline{\alpha}_{\mathbf{y}}}(\bar{\mathbf{x}}, \bar{\mathbf{y}})\| \leq \epsilon.$$

27    As $\epsilon$ is arbitrarily chosen, we have $\|\mathcal{R}_{\alpha_{\mathbf{y}}}(\bar{\mathbf{x}}, \bar{\mathbf{y}})\| \leq 0$ and thus $\bar{\mathbf{y}} \in \hat{S}(\bar{\mathbf{x}})$.

28    Next, as $F$ is continuous at $(\bar{\mathbf{x}}, \bar{\mathbf{y}})$, for any $\epsilon > 0$, there exists $J(\epsilon) > 0$ such that for any $j > J(\epsilon)$,

29    it holds

$$F(\bar{\mathbf{x}}, \bar{\mathbf{y}}) \leq F(\mathbf{x}_j, \mathbf{y}_{i(j)}(\mathbf{x}_j, \mathbf{z}_j)) + \epsilon.$$

30    We define $\hat{\varphi}(\mathbf{x}) := \inf_{\mathbf{y} \in \hat{\mathcal{S}}(\mathbf{x})} F(\mathbf{x}, \mathbf{y})$, then for any $j > J(\epsilon)$ and $\mathbf{x} \in \mathcal{X}$,

$$
\begin{aligned}
\hat{\varphi}(\bar{\mathbf{x}}) &= \inf_{\mathbf{y} \in \hat{\mathcal{S}}(\bar{\mathbf{x}})} F(\bar{\mathbf{x}}, \mathbf{y}) \\
&\leq F(\bar{\mathbf{x}}, \bar{\mathbf{y}}) \\
&\leq F(\mathbf{x}_j, \mathbf{y}_{i(j)}(\mathbf{x}_j, \mathbf{z}_j)) + \epsilon \\
&\leq \max_{1 \leq k \leq j} F(\mathbf{x}_j, \mathbf{y}_k(\mathbf{x}_j, \mathbf{z}_j)) + \epsilon \\
&= \varphi_j(\mathbf{x}_j, \mathbf{z}_j) + \epsilon \\
&\leq \hat{\varphi}(\mathbf{x}) + \epsilon,
\end{aligned}
\tag{1}
$$

31    where the lase inequality follows from Lemma 3.2. By taking $\epsilon \to 0$, we have

$$\hat{\varphi}(\bar{\mathbf{x}}) \leq F(\bar{\mathbf{x}}, \bar{\mathbf{y}}) \leq \hat{\varphi}(\mathbf{x}), \quad \forall \mathbf{x} \in \mathcal{X},$$

32    which implies $\bar{\mathbf{x}} \in \arg\min_{\mathbf{x} \in \mathcal{X}} \hat{\varphi}(\mathbf{x})$ and $(\bar{\mathbf{x}}, \bar{\mathbf{y}}) \in \mathrm{argmin}_{\mathbf{x} \in \mathcal{X}, \mathbf{y} \in \mathcal{Y}} F(\mathbf{x}, \mathbf{y})$, $s.t.$ $\mathbf{y} \in \hat{\mathcal{S}}(\mathbf{x})$. By As-

33    sumption 3.1(5), we have $\bar{\mathbf{y}} \in \mathcal{S}(\mathbf{x})$ and thus $\hat{\varphi}(\bar{\mathbf{x}}) \geq \varphi(\bar{\mathbf{x}})$. Next, since $\hat{\mathcal{S}}(\mathbf{x}) \supset \mathcal{S}(\mathbf{x})$, then $\hat{\varphi}(\mathbf{x}) \leq$

34    $\varphi(\mathbf{x})$ for any $\mathbf{x} \in \mathcal{X}$. Thus we have $\inf_{\mathbf{x} \in \mathcal{X}} \hat{\varphi}(\mathbf{x}) = \inf_{\mathbf{x} \in \mathcal{X}} \varphi(\mathbf{x})$ and $\bar{\mathbf{x}} \in \arg\min_{\mathbf{x} \in \mathcal{X}} \varphi(\mathbf{x})$.

35    We next show that $\inf_{\mathbf{x} \in \mathcal{X}, \mathbf{z} \in \mathcal{Y}} \varphi_K(\mathbf{x}, \mathbf{z}) \to \inf_{\mathbf{x} \in \mathcal{X}} \hat{\varphi}(\mathbf{x}) = \inf_{\mathbf{x} \in \mathcal{X}} \varphi(\mathbf{x})$ as $K \to \infty$. According

36    to Lemma 3.2, for any $\mathbf{x} \in \mathcal{X}$,

$$\inf_{\mathbf{x} \in \mathcal{X}, \mathbf{z} \in \mathcal{Y}} \varphi_K(\mathbf{x}, \mathbf{z}) \leq \hat{\varphi}(\mathbf{x}),$$

37    by taking $K \to \infty$, we have

$$\limsup_{K \to \infty} \left\{ \inf_{\mathbf{x} \in \mathcal{X}, \mathbf{z} \in \mathcal{Y}} \varphi_K(\mathbf{x}, \mathbf{z}) \right\} \leq \hat{\varphi}(\mathbf{x}), \qquad \forall x \in X,$$

38    and thus

$$\limsup_{K \to \infty} \left\{ \inf_{\mathbf{x} \in \mathcal{X}, \mathbf{z} \in \mathcal{Y}} \varphi_K(\mathbf{x}, \mathbf{z}) \right\} \leq \inf_{\mathbf{x} \in \mathcal{X}} \hat{\varphi}(\mathbf{x}).$$

39    So, if $\inf_{\mathbf{x} \in \mathcal{X}, \mathbf{z} \in \mathcal{Y}} \varphi_K(\mathbf{x}, \mathbf{z}) \to \inf_{\mathbf{x} \in \mathcal{X}} \hat{\varphi}(\mathbf{x}) = \inf_{\mathbf{x} \in \mathcal{X}} \varphi(\mathbf{x})$ does not hold, then there exist $\delta > 0$

40    and subsequence $\{(\mathbf{x}_l, \mathbf{z}_l)\}$ of $\{(\mathbf{x}_K, \mathbf{z}_K)\}$ such that

$$\inf_{\mathbf{x} \in \mathcal{X}, \mathbf{z} \in \mathcal{Y}} \varphi_l(\mathbf{x}, \mathbf{z}) = \lim_{l \to \infty} \varphi_l(\mathbf{x}_l, \mathbf{z}_l) < \inf_{\mathbf{x} \in \mathcal{X}} \hat{\varphi}(\mathbf{x}) - \delta, \quad \forall l. \tag{2}$$

41    Since $\mathcal{X}$ is compact, we can assume without loss of generality that $\mathbf{x}_l \to \bar{\mathbf{x}}$ for some $\mathbf{x} \in \mathcal{X}$ by

42    considering a subsequence. Then, as shown in above, we have $\bar{\mathbf{x}} \in \arg\min_{\mathbf{x} \in \mathcal{X}} \hat{\varphi}(\mathbf{x})$. And, by the

43    same arguments for deriving (1), we can show that for any $\epsilon > 0$, there exists $k(\epsilon) > 0$ such that for

44    any $l > k(\epsilon)$, it holds

$$\hat{\varphi}(\bar{\mathbf{x}}) \leq \varphi_l(\mathbf{x}_l, \mathbf{z}_l) + \epsilon.$$

45    By letting $l \to \infty$, $\epsilon \to 0$ and the definition of $\mathbf{x}_l$, we have

$$\inf_{\mathbf{x} \in \mathcal{X}} \hat{\varphi}(\mathbf{x}) = \hat{\varphi}(\bar{\mathbf{x}}) \leq \liminf_{l \to \infty} \left\{ \inf_{\mathbf{x} \in \mathcal{X}, \mathbf{z} \in \mathcal{Y}} \varphi_l(\mathbf{x}, \mathbf{z}) \right\},$$

46    which implies a contradiction to (2). Thus we have $\inf_{\mathbf{x} \in \mathcal{X}, \mathbf{z} \in \mathcal{Y}} \varphi_K(\mathbf{x}, \mathbf{z}) \to \inf_{\mathbf{x} \in \mathcal{X}} \hat{\varphi}(\mathbf{x}) =$

47    $\inf_{\mathbf{x} \in \mathcal{X}} \varphi(\mathbf{x})$ as $K \to \infty$. $\qquad\qquad\qquad\qquad\qquad\qquad\qquad\qquad\qquad\qquad\qquad\qquad\square$

## A.4 Proof of Theorem 3.2

*Proof.* By using the same arguments as in the proof of Theorem 3.1, for any limit point $(\bar{\mathbf{x}}, \bar{\mathbf{z}})$ of the sequence $\{(\mathbf{x}_K, \mathbf{z}_K)\}$, we can find a subsequence $\{(\mathbf{x}_j, \mathbf{z}_j)\}$ of sequence $\{(\mathbf{x}_K, \mathbf{z}_K)\}$ such that $\mathbf{x}_j \to \bar{\mathbf{x}} \in \mathcal{X}$, $\mathbf{z}_j \to \bar{\mathbf{z}} \in \mathcal{Y}$ and $\mathbf{y}_{i(j)}(\mathbf{x}_j, \mathbf{z}_j) \to \bar{\mathbf{y}} \in \mathcal{Y}$ for some $\bar{\mathbf{y}} \in \hat{S}(\bar{\mathbf{x}})$, where $i(K) := \operatorname{argmin}_{0 \leq k \leq K} \|\mathcal{R}_{\alpha_{\mathbf{y}}}(\mathbf{x}, \mathbf{y}_k(\mathbf{x}, \mathbf{z}))\|$.

Next, as $F$ is continuous at $(\bar{\mathbf{x}}, \bar{\mathbf{y}})$, for any $\epsilon > 0$, there exists $J(\epsilon) > 0$ such that for any $j > J(\epsilon)$, it holds

$$F(\bar{\mathbf{x}}, \bar{\mathbf{y}}) \leq F(\mathbf{x}_j, \mathbf{y}_{i(j)}(\mathbf{x}_j, \mathbf{z}_j)) + \epsilon.$$

Then for any $j > J(\epsilon)$,

$$\begin{aligned} F(\bar{\mathbf{x}}, \bar{\mathbf{y}}) &\leq F(\mathbf{x}_j, \mathbf{y}_{i(j)}(\mathbf{x}_j, \mathbf{z}_j)) + \epsilon \\ &\leq \max_{1 \leq k \leq j} F(\mathbf{x}_j, \mathbf{y}_k(\mathbf{x}_j, \mathbf{z}_j)) + \epsilon \\ &= \varphi_j(\mathbf{x}_j, \mathbf{z}_j) + \epsilon. \end{aligned} \tag{3}$$

Next, as $(\mathbf{x}_j, \mathbf{z}_j)$ is a local minimum of $\varphi_j(\mathbf{x}, \mathbf{z})$ with uniform neighborhood modulus $\delta$, it follows

$$\varphi_j(\mathbf{x}_j, \mathbf{z}_j) \leq \varphi_j(\mathbf{x}, \mathbf{z}), \quad \forall (\mathbf{x}, \mathbf{z}) \in \mathbb{B}_\delta(\mathbf{x}_j, \mathbf{z}_j) \cap \mathcal{X} \times \mathcal{Y}.$$

Since $\mathbb{B}_{\delta/2}(\bar{\mathbf{x}}, \bar{\mathbf{z}}) \subseteq \mathbb{B}_{\delta/2 + \|(\mathbf{x}_j, \mathbf{z}_j) - (\bar{\mathbf{x}}, \bar{\mathbf{z}})\|}(\bar{\mathbf{x}}, \bar{\mathbf{z}}) \subseteq \mathbb{B}_\delta(\mathbf{x}_j, \mathbf{z}_j)$ when $\|(\mathbf{x}_j, \mathbf{z}_j) - (\bar{\mathbf{x}}, \bar{\mathbf{z}})\| < \delta/2$, we have that there exists $J(\delta) > 0$ such that whenever $j > J(\delta)$, for any $(\mathbf{x}, \mathbf{z}) \in \mathbb{B}_{\delta/2}(\bar{\mathbf{x}}, \bar{\mathbf{z}}) \cap \mathcal{X} \times \mathcal{Y}$,

$$\varphi_j(\mathbf{x}_j, \mathbf{z}_j) \leq \varphi_j(\mathbf{x}, \mathbf{z}).$$

Then, applying the same arguments as in the proof of Lemma 3.2 yields that whenever $j > J(\delta)$,

$$\varphi_j(\mathbf{x}_j, \mathbf{z}_j) \leq F(\mathbf{x}, \mathbf{z}),$$

for any $(\mathbf{x}, \mathbf{z}) \in \mathbb{B}_{\tilde{\delta}}(\bar{\mathbf{x}}, \bar{\mathbf{z}}) \cap \{\mathbf{x} \in \mathcal{X}, \mathbf{z} \in \mathcal{Y} \mid \mathbf{z} \in \hat{S}(\mathbf{x})\}$ with $\tilde{\delta} = \delta/2$. Combining with (3) and taking $j \to \infty$, $\epsilon \to 0$ gives the conclusion. $\qquad\square$

# B  Proof of results in Section 3.2

**Lemma B.1.** *[3, Lemma 1] Denote $f^*(\mathbf{x}) := \min_{\mathbf{y}} f(x, y)$. If $f(\mathbf{x}, \mathbf{y})$ is continuous on $\mathcal{X} \times \mathbb{R}^m$, then $f^*(\mathbf{x})$ is upper semi-continuous on $\mathcal{X}$.*

**Lemma B.2.** *Assume that $\mathbf{y}_k(\mathbf{x}, \mathbf{z})$ satisfies $\mathbf{y}_k(\mathbf{x}, \mathbf{z}) = \mathbf{z}$ for any $\mathbf{z} \in S(\mathbf{x})$, $\mathbf{x} \in \mathcal{X}$ and $k \geq 0$. Let $(\mathbf{x}_K, \mathbf{z}_K) \in \operatorname{argmin}_{\mathbf{x} \in \mathcal{X}, \mathbf{z} \in \mathcal{Y}} \phi_K(\mathbf{x}, \mathbf{z}) := F(\mathbf{x}, \mathbf{y}_K(\mathbf{x}, \mathbf{z}))$, then*

$$\phi_K(\mathbf{x}_K, \mathbf{z}_K) \leq \varphi(\mathbf{x}), \quad \forall \mathbf{x} \in \mathcal{X}.$$

*Proof.* For any $\mathbf{x} \in \mathcal{X}$, and any $\epsilon > 0$, there exists $\mathbf{y}_\epsilon \in S(\mathbf{x})$ such that $F(\mathbf{x}, \mathbf{y}_\epsilon) \leq \varphi(\mathbf{x}) + \epsilon$. As $\mathbf{y}_\epsilon \in S(\mathbf{x})$, then by assumption that $\mathbf{y}_k(\mathbf{x}, \mathbf{y}_\epsilon) = \mathbf{y}_\epsilon$ for any $k \geq 0$. Since $\mathbf{y}_\epsilon \in \mathcal{Y}$, we have

$$\phi_K(\mathbf{x}_K, \mathbf{z}_K) \leq \phi_K(\mathbf{x}, \mathbf{y}_\epsilon) = F(\mathbf{x}, \mathbf{y}^k(\mathbf{x}, \mathbf{y}_\epsilon)) = F(\mathbf{x}, \mathbf{y}_\epsilon) \leq \varphi(\mathbf{x}) + \epsilon.$$

The conclusion follows by letting $\epsilon \to 0$ in above inequality. $\qquad\square$

## B.1  Proof of Theorem 3.3

*Proof.* For any limit point $\bar{\mathbf{x}}$ of the sequence $\{\mathbf{x}_K\}$, let $\{\mathbf{x}_l\}$ be a subsequence of $\{\mathbf{x}_K\}$ such that $\mathbf{x}_l \to \bar{\mathbf{x}} \in \mathcal{X}$. As $\{\mathbf{y}_K(\mathbf{x}_K, \mathbf{z}_K)\} \subset \mathcal{Y}$ is bounded, we can have a subsequence $\{\mathbf{x}_j\}$ of $\{\mathbf{x}_l\}$ satisfying $\mathbf{y}_j(\mathbf{x}_j, \mathbf{z}_j) \to \bar{\mathbf{y}}$ for some $\bar{\mathbf{y}} \in \mathcal{Y}$. When the condition (a) holds, for any $\epsilon > 0$, there exists $J(\epsilon) > 0$ such that for any $j > J(\epsilon)$, we have

$$f(\mathbf{x}_j, \mathbf{y}_j(\mathbf{x}_j, \mathbf{z}_j)) - f^*(\mathbf{x}_j) \leq \epsilon.$$

75    By letting $j \to \infty$, and since $f$ is continuous and $f^*(x)$ is upper semi-continuous on $\mathcal{X}$ from Lemma
76    B.1, we have

$$f(\bar{\mathbf{x}}, \bar{\mathbf{y}}) - f^*(\bar{\mathbf{x}}) \le \epsilon.$$

77    As $\epsilon$ is arbitrarily chosen, we have $f(\bar{\mathbf{x}}, \bar{\mathbf{y}}) - f^*(\bar{\mathbf{x}}) \le 0$ and thus $\bar{\mathbf{y}} \in S(\bar{\mathbf{x}})$.

78    On the other hand, if $\mathbf{y}_k(\mathbf{x}, \mathbf{z})$ satisfies condition (b). For any $\epsilon > 0$, there exists $J(\epsilon) > 0$ such that
79    for any $j > J(\epsilon)$, we have

$$\|\mathcal{R}_\alpha(\mathbf{x}_j, \mathbf{y}_j(\mathbf{x}_j, \mathbf{z}_j))\| \le \epsilon.$$

80    By letting $j \to \infty$, and since $\mathcal{R}_\alpha$ is continuous, we have

$$\|\mathcal{R}_\alpha(\bar{\mathbf{x}}, \bar{\mathbf{y}})\| \le \epsilon.$$

81    As $\epsilon$ is arbitrarily chosen, we have $\|\mathcal{R}_\alpha(\bar{\mathbf{x}}, \bar{\mathbf{y}})\| \le 0$ and thus $\bar{\mathbf{y}} \in S(\bar{\mathbf{x}})$.

82    Next, as $F$ is continuous at $(\bar{\mathbf{x}}, \bar{\mathbf{y}})$, for any $\epsilon > 0$, there exists $J(\epsilon) > 0$ such that for any $j > J(\epsilon)$,
83    it holds

$$F(\bar{\mathbf{x}}, \bar{\mathbf{y}}) \le F(\mathbf{x}_j, \mathbf{y}_j(\mathbf{x}_j, \mathbf{z}_j)) + \epsilon.$$

84    Then, we have, for any $j > J(\epsilon)$ and $\mathbf{x} \in \mathcal{X}$,

$$
\begin{aligned}
\varphi(\bar{\mathbf{x}}) = \inf_{\mathbf{y} \in S(\bar{\mathbf{x}})} F(\bar{\mathbf{x}}, \mathbf{y}) \\
\le F(\bar{\mathbf{x}}, \bar{\mathbf{y}}) \\
\le F(\mathbf{x}_j, \mathbf{y}_j(\mathbf{x}_j, \mathbf{z}_j)) + \epsilon \\
= \phi_j(\mathbf{x}_j, \mathbf{z}_j) + \epsilon \\
\le \varphi(\mathbf{x}) + \epsilon,
\end{aligned}
\tag{4}
$$

85    where the lase inequality follows from Lemma B.2. By taking $\epsilon \to 0$, we have

$$\varphi(\bar{\mathbf{x}}) \le \varphi(\mathbf{x}), \quad \forall \mathbf{x} \in \mathcal{X},$$

86    which implies $\bar{\mathbf{x}} \in \arg\min_{\mathbf{x} \in \mathcal{X}} \varphi(\mathbf{x})$.

87    We next show that $\inf_{\mathbf{x} \in \mathcal{X}, \mathbf{z} \in \mathcal{Y}} \phi_K(\mathbf{x}, \mathbf{z}) \to \inf_{\mathbf{x} \in \mathcal{X}} \varphi(\mathbf{x})$ as $K \to \infty$. According to Lemma B.2,
88    for any $\mathbf{x} \in \mathcal{X}$,

$$\inf_{\mathbf{x} \in \mathcal{X}, \mathbf{z} \in \mathcal{Y}} \phi_K(\mathbf{x}, \mathbf{z}) \le \varphi(\mathbf{x}),$$

89    by taking $K \to \infty$, we have

$$\limsup_{K \to \infty} \left\{ \inf_{\mathbf{x} \in \mathcal{X}, \mathbf{z} \in \mathcal{Y}} \phi_K(\mathbf{x}, \mathbf{z}) \right\} \le \varphi(\mathbf{x}), \qquad \forall \mathbf{x} \in \mathcal{X},$$

90    and thus

$$\limsup_{K \to \infty} \left\{ \inf_{\mathbf{x} \in \mathcal{X}, \mathbf{z} \in \mathcal{Y}} \phi_K(\mathbf{x}, \mathbf{z}) \right\} \le \inf_{\mathbf{x} \in \mathcal{X}} \varphi(\mathbf{x}).$$

91    So, if $\inf_{\mathbf{x} \in \mathcal{X}, \mathbf{z} \in \mathcal{Y}} \phi_K(\mathbf{x}, \mathbf{z}) \to \inf_{\mathbf{x} \in \mathcal{X}} \varphi(\mathbf{x})$ does not hold, then there exist $\delta > 0$ and subsequence
92    $\{(\mathbf{x}_l, \mathbf{z}_l)\}$ of $\{(\mathbf{x}_K, \mathbf{z}_k)\}$ such that

$$\inf_{\mathbf{x} \in \mathcal{X}, \mathbf{z} \in \mathcal{Y}} \phi_l(\mathbf{x}, \mathbf{z}) = \lim_{l \to \infty} \phi_l(\mathbf{x}_l, \mathbf{z}_l) < \inf_{\mathbf{x} \in \mathcal{X}} \varphi(\mathbf{x}) - \delta, \quad \forall l. \tag{5}$$

93    Since $\mathcal{X}$ is compact, we can assume without loss of generality that $\mathbf{x}_l \to \bar{\mathbf{x}}$ for some $\mathbf{x} \in \mathcal{X}$ by
94    considering a subsequence. Then, as shown in above, we have $\bar{\mathbf{x}} \in \arg\min_{\mathbf{x} \in \mathcal{X}} \varphi(\mathbf{x})$. And, by the
95    same arguments for deriving (4), we can show that for any $\epsilon > 0$, there exists $k(\epsilon) > 0$ such that for
96    any $l > k(\epsilon)$, it holds

$$\varphi(\bar{\mathbf{x}}) \le \phi_l(\mathbf{x}_l, \mathbf{z}_l) + \epsilon.$$

97    By letting $l \to \infty$, $\epsilon \to 0$ and the definition of $\mathbf{x}_l$, we have

$$\inf_{\mathbf{x} \in \mathcal{X}} \varphi(\mathbf{x}) = \varphi(\bar{\mathbf{x}}) \le \liminf_{l \to \infty} \left\{ \inf_{\mathbf{x} \in \mathcal{X}, \mathbf{z} \in \mathcal{Y}} \phi_l(\mathbf{x}, \mathbf{z}) \right\},$$

98    which implies a contradiction to (5). Thus we have $\inf_{\mathbf{x} \in \mathcal{X}, \mathbf{z} \in \mathcal{Y}} \phi_K(\mathbf{x}, \mathbf{z}) \to \inf_{\mathbf{x} \in \mathcal{X}} \varphi(\mathbf{x})$ as $K \to$
99    $\infty$. $\qquad\square$

## B.2 Proof of Theorem 3.4

*Proof.* According to [1, Theorem 10.34], when $f(\mathbf{x}, \cdot)$ is convex and $L_f$-smooth for any $\mathbf{x} \in \mathcal{X}$, and $\alpha = \frac{1}{L_f}$, $\{\mathbf{y}_k(\mathbf{x}, \mathbf{z})\}$ admits the following property,

$$f(\mathbf{x}, \mathbf{y}_K(\mathbf{x}, \mathbf{z})) - f^*(\mathbf{x}) \leq \frac{2L_f \text{dist}(\mathbf{y}_0(\mathbf{x}, \mathbf{z}), \mathcal{S}(\mathbf{x}))}{(k+1)^2} = \frac{2L_f \text{dist}(\mathbf{z}, \mathcal{S}(\mathbf{x}))}{(k+1)^2},$$

where $\text{dist}(\mathbf{z}, S(\mathbf{x}))$ denotes the distance from $\mathbf{z}$ to the set $S(\mathbf{x})$. Since $\mathcal{X}$ and $\mathcal{Y}$ are both compact sets, then there exists $M > 0$ such that $\text{dist}(\mathbf{z}, \mathcal{S}(\mathbf{x})) \leq M$ for $(\mathbf{x}, \mathbf{z}) \in \mathcal{X} \times \mathcal{Y}$. Then we can easily obtained from the above lemma that $\{y_k(x, z)\}$ satisfies condition (a) in Theorem 3.3. Next, $\mathbf{y}_k(\mathbf{x}, \mathbf{z}) \in \mathcal{Y}$ follows from the update formula of $\mathbf{y}_k$ immediately. And when $\mathbf{u}_0(\mathbf{x}, \mathbf{z}) = \mathbf{y}_0(\mathbf{x}, \mathbf{z}) = \mathbf{z} \in \mathcal{S}(\mathbf{x})$, it can be easily verified that $\mathbf{u}_k(\mathbf{x}, \mathbf{z}) = \mathbf{y}_k(\mathbf{x}, \mathbf{z}) = \mathbf{z}$ for any $k \geq 0$. Thus $\{\mathbf{y}_k(\mathbf{x}, \mathbf{z})\}$ satisfies all the assumptions required by Theorem 3.3. $\qquad\square$

# C Experiments

Our experiments were conducted on a PC with Intel Core i9-10900KF CPU (3.70GHz), 128GB RAM, two NVIDIA GeForce RTX 3090 24GB GPUs, and the platform is 64-bit Ubuntu 18.04.5 LTS.

## C.1 Non-convex Numerical Example

For the non-convex BLO problem within the text, we follow the parameter settings in Table 1. The EG methods and our IAPTT-GM follow the general setting of hyperparameters, and IG methods follow the instruction of specific hyperparameters.

Note that we adopt SGD optimizer for updating UL variables $\mathbf{x}$ and initialization auxiliary $\mathbf{z}$. $\mathcal{T}$ denotes the inner iterations number for IG methods, e.g., LS and NS. $\mu$ denotes the ratio between UL and LL objectives when aggregating the LL and UL gradients for BDA [3], $\mu \in (0, 1)$ .

Table 1: Values for hyper parameters of nonconvex numerical examples.

| General setting | Value |
| --- | --- |
| Outer loop | 500 |
| Inner loop | 40 |
| Learning rate | 0.0005 |
| Meta learning rate | 0.1 |
| Specific hyperparameter | Value |
| Inner iteration $\mathcal{T}$ | 40 |
| Ratio $\mu$ | 0.4 |

## C.2 Few-Shot Classification

**Datasets**. We choose two well-known benchmarks constructed from the ILSVRC-12 dataset named miniImageNet [6] and TieredImageNet [5]. The miniImgaenet consists of 100 selected classes, and each of the class contains 600 downsampled images of size $84 \times 84$. The whole dataset is divided into three disjoint subsets: 64 classes for training, 16 for validation, and 20 for testing. The tieredImageNet is a larger subset with 608 classes, including 779,165 images of the same size in total. These classes are split into 20, 6, 8 categories like miniImageNet, resulting in 351, 97,

127 160 classes as training, validation, testing set, respectively. Few shot classification task on the
128 tieredImageNet is more challenging due to its dissimilarity between training and testing sets.

129 **Network Structures**. We employ the ConvNet-4 [2] and ResNet-12 [4] network structures, which
130 are commonly used in few shot classification tasks. ConvNet-4 is a 4-layer convolutional neural
131 network with $k$ filters followed by batch normalization, non-linearity, and max-pooling operation.
132 ResNet-12 consists of 4 residual blocks followed by 2 convolutional layers, and each block has
133 three repeated groups, including $\{3 \times 3$ convolution with $k$ filters, batch normalization, activation
134 function$\}$. Both of the network structures adopt the fully connected layer with softmax function as
135 the baseline classifier.

136 We adopt Adam for updating UL variables $\mathbf{x}$ and initialization auxiliary $\mathbf{z}$ in our method and UL
variable $\mathbf{x}$ in other methods for fair comparison. Related hyperparameters are stated in Table 2.

Table 2: Values for hyperparameters of few shot classification.

| General setting | ConvNet-4 | ResNet-12 |
| --- | --- | --- |
| Outer loop | 80000 | 80000 |
| Inner loop | 10 | 10 |
| Learning rate | 0.1 | 0.1 |
| Meta learning rate | 0.001 | 0.001 |
| Meta batch size | 4 | 2 |
| Hidden size | 32 | 48 |
| Ratio $\mu$ | 0.4 | 0.4 |

137

## C.3 Data Hyper-Cleaning

138

139 We use the subsets of MNIST dataset and more challenging FashionMNIST dataset for training. The
140 MNIST database includes handwritten digits (0 through 9), which is widely used for classification
141 tasks. The FashionMNIST contains different categories of clothing, and serves as a direct drop-in
142 replacement for the original MNIST dataset. The subsets are randomly split to three disjoint subsets,
143 which contain 5000, 5000, 10000 examples, respectively. We adopt Adam for updating variables $\mathbf{x}$
144 and $\mathbf{z}$ in our method and UL variables $\mathbf{x}$ in other methods for fair comparison. The values of hyper
parameters are listed in Table 3.

Table 3: Values for hyperparameters of data hyper-cleaning.

| General setting | Value |
| --- | --- |
| Outer loop | 3000 |
| Inner loop | 50 |
| Learning rate | 0.03 |
| Meta learning rate | 0.01 |
| Specific hyperparameter | Value |
| Inner iteration $\mathcal{T}$ | 50 |
| Ratio $\mu$ | 0.4 |

145

## C.4 Details for Evaluation of IA-GM (A)

We conduct the acceleration experiments following the parameters setting given in Table 4. Note that we adopt SGD for updating variables $\mathbf{x}$ and $\mathbf{z}$.

Table 4: Values for Hyper parameters of convex numerical examples.

| General setting | Value |
|---|---|
| Outer loop | 1000 |
| Inner loop | 20 |
| Learning rate | 0.15 |
| Meta learning rate | 0.005 |