# OpenReview forum: "Towards Gradient-based Bilevel Optimization with Non-convex Followers and Beyond"
_NeurIPS.cc/2021/Conference — NeurIPS 2021 Spotlight_

### Official Review · Reviewer_b25m · 2021-07-13

**Rating:** 7
**Confidence:** 3

**Summary:**

This paper presents a first-order algorithm for solving bilevel optimization problems where the lower-level problem is non-convex. The proposed algorithm, named as IAPTT-GM, in my understanding, is roughly alternating projected gradient descent in the upper level and lower level variables, combined with two key ideas, namely initialization auxiliary and pessimistic trajectory truncation.

This paper provides theoretical analysis of IAPTT-GM, supported by sufficient numerical experiments, both on toy problems and real datasets.

This paper is well motivated and well written. The theoretical part is clean and concise, while the numerical experiments are well conducted, together with texts to describe the main takeaways.

In general, I like this paper and want to see it published. However, I am a bit skeptical about the theory. I may downgrade my score if there are severe theoretical issues. See main review.

**Limitations And Societal Impact:**

See main review.

**Main Review:**

The biggest confusion for me is, Theorem 3.1 seems to be so strong that it violates NP-hardness results of generic non-convex problems.

The bilevel optimization problem considered in this paper, neither has a convex upper level problem, nor has a convex lower level problem. Therefore, to me the problem is obviously non-convex and NP-hard, not to mention about the extra difficulty imposed by the bilevel structure. In addition, the proposed algorithm is a first order algorithm.  However, Theorem 3.1 claims that the algorithm can find a "solution" to the upper level problem.  How can this be possible? I wonder what does the term "solution" mean in this context, does it mean a first-order stationary point, or does it mean the "globally optimal solution"?  Intuitively, the best thing one can hope for first order methods is convergence to a first-order stationary point. So this result is a bit too strong for me. Am I missing some strong assumptions of Theorem 3.1?  I'd like the authors to clarify this point. I will also read the reviews. from other reviewers to see if they have similar concerns.

The other major comment is, how efficient is the proposed IAPTT-GM algorithm, compared with other methods? From the description of Algorithm 1, it seems many inner and outer iterations are required. Could the authors, in the experimental section, comment more on the runtime and efficiency of IAPTT-GM when compared with other baselines?

Bilevel programming is an active area of research in the mathematical programming community. I also wonder if the authors could draw connections to literature in that field when presenting related works. In particular, the value function approach seems to be a preferred approach for solving such problems. For example, see [Ref1] and references therein (the authors of [Ref1] seem to have made great efforts along this vein).

[Ref1]: Ye, Jane J., Xiaoming Yuan, Shangzhi Zeng, and Jin Zhang. "Difference of convex algorithms for bilevel programs with applications in hyperparameter selection." arXiv preprint arXiv:2102.09006 (2021).

**Time Spent Reviewing:**

2

---

> ### Author Response · Authors · 2021-08-10
> **Response to Reviewer #4**
>
> We thank the reviewer for the positive reviews and insightful questions. We address your comments below.
>
> **Q1.** “...to me the problem is obviously non-convex and NP-hard...However, Theorem 3.1 claims that the algorithm can find a ‘solution’ to the upper level problem...what does the term ‘solution’ mean in this context...”
>
> **R1.** First of all, we agree with the reviewer that the bilevel problem  in Eq. (1)  is indeed non-convex and NP-hard. While our proposed method is in essence a sequential unconstrained minimization type method. Specifically, by unrolling the lower-level problem with projected gradient scheme, we obtain a series of unconstrained approximated subproblems “$\min \varphi_K(\mathbf{x},\mathbf{z})$”. These approximation problems are still non-convex and NP-hard. Fortunately, they are readily trackable by celebrated optimization methods (e.g., autodiff) since the hierarchical structure is no longer there.  In summary, instead of solving the original NP-hard problem with bilevel structure, we design an algorithm to solve a series of approximation problems which are numerically trackable.
>
> As for the solution concern, we would like to emphasize that, our submission paper is essentially a sequential unconstrained minimization type method. In fact, the mainstream of gradient-based bilevel methods belongs to this class. In particular, by unrolling the lower-level and upper level objectives with gradient method scheme, one obtains a series of unconstrained approximated subproblems. The approximation quality is typically justified in theory from two aspects. If a series of global solutions of approximation subproblems are found, then a global solution of the original bilevel problem can be approximatedly achieved. Alternatively, if a series of local solutions of approximation subproblems are found, then a local solution of the original bilevel problem can be approximatedly achieved.
>
> The approximation quality is our main contribution. Particularly, in our submission paper, we actually construct a series of unconstrained approximated subproblems under the weakest structure assumption among the literature, i.e., the lower level non-convexity. We have proved the approximation quality in our paper from the global solution aspect. To our best knowledge, we establish the first strict convergence guarantee for gradient-based method on bilevel with non-convex lower level. For this purpose, technically, we propose IA and PTT, two new mechanisms to efficiently handle the lower level non-convexity.
>
> Please also refer to response to Reviewer 1 for related issue.
>
> **Q2.** “...it seems many inner and outer iterations are required. Could the authors, in the experimental section, comment more on the runtime and efficiency of IAPTT-GM when compared with other baselines?...”
>
> **R2.** Thanks for the question. In fact, IA and PTT are the two new components in our gradient-based BLO algorithms. We  emphasize that by introducing these modules, we have obtained a new method with strict convergence guarantee for BLOs with non-convex follower structure. Please notice that such challenging BLO problems have never been properly addressed by existing approaches and this should be the main breakthroughs of our work.
>
>
> With such IA and PTT modules, we have that the overall IAPTT-GM scheme is efficient in practice. In fact, compared with existing gradient-based BLOs (e.g., RHG and BDA), the only additional iterations in IAPTT-GM is the $\mathbf{z}$-related subproblem required by IA. While our PTT strategy can significantly reduce the length of iteration trajectory for back-propagation. Numerical experiments can also verify this justification. That is, we have compared the computational memory and running time of IAPTT-GM with other mainstream approaches on our numerical example (see R1 to Reviewer 2 above). We observe that IAPTT-GM performs better than many existing approaches and only slightly slower than these simple iteration schemes. Please also notice that the theoretical properties of these simple methods are much weaker than ours.
>
> We will add necessary discussions on these issues in our revised version.
>
>
> **Q3.** “...if the authors could draw connections to literature in that field when presenting related works. In particular, the value function approach seems to be a preferred approach for solving such problems...”
>
> **R3.** Thanks for pointing out related works for us. Indeed, the value function approach was introduced to tackle the optimality condition for bilevel problem. However, numerical methods based on this technique are quite limited in the literature, because the value-function generally does not admit an explicit form. In fact, the value function is always nonsmooth, non-convex and with jumps.
>
> The reference mentioned by the reviewer actually studies bilevel problem with a very special underlying structure. That work considers the case where the lower-level is jointly convex with respect to both the upper level and lower level variables. Thanks to this structure, the bilevel can be equivalently reformulated into a difference-of-convex program, which is numerically solvable. While in our paper, the setting is fairly general, i.e., non-convex lower level. We cannot expect such a good difference-of-convex reformulation by using the value function approach.
>
> We will add more discussions on these related works in our updated version.

---

> > ### Comment · Reviewer_b25m · 2021-08-17
> > **Thank you for the clarifications**
> >
> > I have read other reviews and the authors' comments. I'd like to keep my original score as an accept.

---

### Official Review · Reviewer_BsfL · 2021-07-13

**Rating:** 7
**Confidence:** 4

**Summary:**

This work proposes a truncated unrolling type method for solving a bilevel optimization problem with non-convex lower-level problem and the convergence of the proposed method is also shown. Compared to the results in the existing literature, the convergence analysis in this work does not require any convexity assumptions on both upper level and lower-level objective functions. Efficiency of the proposed method is shown in the numerical experiments of artificial problems and real application problems.

**Ethical Concerns:**

N.A.

**Ethics Review Area:**

["I don’t know"]

**Limitations And Societal Impact:**

Yes.

**Main Review:**

This work proposes a truncated unrolling type method for solving a bilevel optimization problem with non-convex lower-level problem and the convergence of the proposed method is also shown. Compared to the results in the existing literature, the convergence analysis in this work does not require any convexity assumptions on both upper level and lower-level objective functions. Efficiency of the proposed method is shown in the numerical experiments of artificial problems and real application problems.

I noticed that two interesting ideas, namely, Initialization Auxiliary and Pessimistic Trajectory Truncation, are applied in the designing of the proposed method. Initialization Auxiliary is a well-used technique in meta-learning, but the Pessimistic Trajectory Truncation technique is quite new. Thanks to the Pessimistic Trajectory Truncation technique, the proposed method admits a convergence guarantee, meanwhile, the computation cost in the numerical implementation is reduced. I think that such Pessimistic Trajectory Truncation technique may bring something new to other bilevel optimization algorithm designs.

Cons:
1. The convergence result of the proposed method is based on the sequence x_K being global minimum of the approximated subproblems. Since the approximation is just with respect to function values, I understand that it is usually too ambitious to establish the stationary points convergence in such a general case (nonconvex LL) considered in this paper. However, since the subproblems are always nonconvex, this referee would like to ask is it possible to derive convergence analysis for the case where x_K are local minimum.
2. I understand that the LL compact set constraint Y is imposed for theoretical convergence analysis. The authors should clarify that how to set and deal with this extra Y in numerical experiments.
3. In section 3.2, the authors embed the Nesterov’s acceleration dynamics into their algorithm. It seems that the convergence analysis does not rely on any particular features of the Nesterov’s acceleration dynamics. This referee feels that this part can be generalized to other acceleration dynamics.
4. The LL subproblem is now set to be differentiable. This referee wonder if the general non-differentiable case can be considered to capture more real applications.
5. In the numerical experiments, the authors conduct a comparison for the proposed methods with and without Pessimistic Trajectory Truncation on an artificial problem. Empirically, the Pessimistic Trajectory Truncation technique brings a great improvement. I was wondering whether such an improvement also exists for real application problems.

**Final Comment (Post Rebuttal)**

I acknowledge that I read and other reviewers' comments and authors' responses. I would like to keep my score as most concerns of mine have been well addressed.


**Time Spent Reviewing:**

5

---

> ### Author Response · Authors · 2021-08-10
> **Response to Reviewer #3**
>
> We thank the reviewer for recognizing the novelty of this work. Responses to the questions are below.
>
> **Q1.** “...is it possible to derive convergence analysis for the case where x\_K are local minimum...”
>
> **R1.** Thanks for the question. We should point out that our theoretical analysis framework is indeed flexible enough to derive convergence results for the case $\mathbf{x}_K$ are local minimum. That is, we just need to extend Lemma 3.2 to a local version, and accordingly, establish the convergence result of local minimums to certain kind of local solution of the bilevel problem. Please also see the reply to Reviewer 1 above for more details.
>
> **Q2.** “...The authors should clarify that how to set and deal with this extra Y in numerical experiments...”
>
> **R2.** Thanks for the suggestion. Indeed, the set constraint $\mathcal{Y}$ is only used to guarantee the completeness of our theoretical analysis. Thus in application scenarios, we can just consider the constraint as an extra large set, so that all the varietals are automatically in this feasible set. In this way, it is natural to ignore the projection operation in practical computations. We will add necessary statements to clarify this issue in our revised manuscript.
>
> **Q3.** “...In section 3.2, the authors embed the Nesterov’s acceleration dynamics into their algorithm... this part can be generalized to other acceleration dynamics...”
>
> **R3.** Thanks for the comment. The answer is positive. Indeed, Nesterov's acceleration dynamics can be replaced by any algorithms satisfying either one of the two conditions in Theorem 3.2. So many widely-used first-order methods satisfy such condition.
>
> **Q4.** “...if the general non-differentiable case can be considered to capture more real applications...”
>
> **R4.** The answer of this question is also positive. In particular, our method can be easily extended to the case where the LL subproblem is in form of $f + g$, in which $f$ is smooth and $g$ is nonsmooth by replacing the projected gradient dynamics in the lower-level by the proximal gradient dynamics. The theoretical results and proofs are almost the same. But one problem for this setting is that the nonsmoothness may bring difficulties in back-propagation for calculating the hyper-gradient, because we need to calculate the gradient of the proximal operator.
>
> **Q5.** “...the authors conduct a comparison for the proposed methods with and without Pessimistic Trajectory Truncation on an artificial problem...whether such an improvement also exists for real application problems...”
>
> **R5.** Thanks for the question. PTT actually provides us a technique to automatically determine the trajectory for back-propagation. In fact, it not only improves the numerical results, but also has its own advantages for real applications. We have compared our method w/ and w/o PTT on few-shot learning task and experimentally observed that PTT can improve the performance by around 2\%.

---

> > ### Comment · Reviewer_BsfL · 2021-08-12
> > **Thank you for your clarifications.**
> >
> > I acknowledge that I read and other reviewers' comments and authors' responses. I would like to keep my score as most concerns of mine have been well addressed.

---

### Official Review · Reviewer_pFSa · 2021-07-16

**Rating:** 7
**Confidence:** 3

**Summary:**

The paper introduces a gradient-based bilevel optimization algorithm called IAPTT-GM that does not require the non-convex inner-level assumption. Specifically, the authors propose two methods: (1) initialization auxiliary that guides the optimization dynamics and (2) pessimistic truncation operation to build theoretical convergence. The authors theoretically show convergence of IAPTT-GM that does not assume the non-convexity of the inner-level problem and empirically show that the proposed method has improved performance on few-shot classification and hyper cleaning compared to previous approaches.


**Limitations And Societal Impact:**

While the authors did not address the potential negative social impact of their work, I believe that it is not necessary for this submission.

**Main Review:**

1. Originality
 * The task of solving bilevel optimization that does not require the lower-level singleton (LLS) assumption has been explored in previous works (e.g. [1]).
*  However, I am not aware of the pessimistic trajectory truncated method to solve this particular issue and I believe that the algorithms & theoretical analysis presented in the paper are interesting and novel.

2. Quality
* The submission is technically sound. I checked the derivation and proofs and they look correct.
* The potential limitations of the proposed approach are not addressed in the paper. For example, what would be the computational & memory overhead compared to previous approaches?

3. Clarity:
 * The manuscript is overall well written and easy to understand. However, the paper uses several abbreviations that are not properly defined (e.g. UL, LL, LLS, LLC). It would be much helpful to properly define these abbreviations. Having a table describing these abbreviations would be helpful.
* Moreover, it would be helpful to summarize all mathematical notation used in the paper. Some notations have quite different interpretations with a subscript which makes proofs hard to read.

4. Significance:
 * Gradient-based bilevel optimization that does not require the lower-level singleton (LLS) assumption is a recently emerging topic in the literature and I believe that result is important. It is relevant to the Neurips community.

5. Additional Comments:
* I don't exactly understand line 150: "The pessimistic max operation always results in a favorable trajectory truncation smaller than K". What would be the reason behind this statement?

6. Minor Comments:
* For gradient-based methods, I feel like the argmin formulation in equation 2 is weird. Can the rational reaction set S(x) map a solution of the local minima? Shouldn't the version that incorporates local minima (e.g. [2]) be more correct?
* In line 122, relevant references on convergence theory of non-convex first-order optimization would be helpful.
* In line 136, the footnote should be inside the comma.
* Introducing variables like T, K would be helpful in describing algorithm 1.
* I feel like showing the baseline algorithm next to algorithm 1 would help readers to better understand contributions made in this paper.

[1] Liu, Risheng, et al. "A generic first-order algorithmic framework for bi-level programming beyond lower-level singleton." International Conference on Machine Learning. PMLR, 2020.

[2] Fiez, Tanner, Benjamin Chasnov, and Lillian J. Ratliff. "Convergence of learning dynamics in stackelberg games." arXiv preprint arXiv:1906.01217 (2019).

------
Thank you for your clarifications. I acknowledge that I read and other reviewers' comments and authors' responses.

I agree that properly defining abbreviations and showing the baseline algorithm would further add clarity to the paper (as well as the new assumption suggested by the authors). Given that these modifications would be made in the manuscript, I will increase my score to 7 (from 6).

**Time Spent Reviewing:**

5

---

> ### Author Response · Authors · 2021-08-10
> **Response to Reviewer #2**
>
> We thank the reviewer for the positive and detailed review as well as the suggestions for improvement. Answers to specific points are below:
>
> **Q1.** “...what would be the computational & memory overhead compared to previous approaches?...”
>
> **R1.** We first emphasize that our main scope of this work is to develop new algorithm to solve BLOs with non-convex follower structure and no exiting works can provide strict convergence guarantee in this optimization scenario. To address this issue, we introduced IA and PTT to control the iteration process and guarantee the convergence of the proposed scheme.
>
> Then we clarify that even with these two additional components, our method is still efficient. This is because that although IA introduces additional variables and iterations, PTT can choose a small truncation number, thus shortens the back-propagation trajectory for computing the UL gradient (see Fig. 2). For example, in the numerical example, the memory required by our IAPTT-GM (10.153 GB) is less than NS (10.387GB), LS (10.426GB), and BDA (10.154GB) and the same as that for RHG (10.153GB). The corresponding running time are IAPTT (10.835s), NS (10.815s), LS (5.120s), BDA (16.800s) and RHG (9.990s). But please notice that the theoretical properties of IAPTT-GM is much better than these existing approaches (see Fig.~1 for numerical verification).
>
> **Q2.** “...It would be much helpful to properly define these abbreviations...”
>
> **R2.** Thanks for the suggestion. As for UL (and LL), it is the abbreviation of “Upper-Level” (and “Lower-Level”). LLS and LLC are abbreviations of “Lower Level Singleton” and “Lower Level Convexity”, which have been defined in lines 36 and 4, respectively. In Alg.~1, $K$ and $T$ denote the numbers of inner and outer iterations, respectively. We will properly define these abbreviations in our revision.
>
> Besides, we appreciate the suggestion for showing the baseline algorithm and will add this material in the latter version. We also thank the reviewer for suggestions of relevant references, and they will be properly cited for completeness.
>
> **Q3.** “...I don’t exactly understand line 150, ‘The pessimistic max operation always results in a favorable trajectory truncation smaller than K...”
>
> **R3.** Thanks for the question. In fact, $K$ is the length of the whole forward-propagation trajectory. According to the definition of $\bar{k}$ given in Alg.~1 (step 9), we have that $K$ actually provides us an upper bound for $\bar{k}$. That's the reason why we say that the trajectory truncation (i.e., $\bar{k}$) is smaller than $K$.
>
> **Q4.** “...Can the rational reaction set S(x) map a solution of the local minima? Shouldn’t the version that incorporates local minima (e.g. [2]) be more correct? ...”
>
> **R4.** Thanks for the questions. We first emphasize that the BLO formulation in Eq. (1) follows the classical setting,  where $\mathcal{S}(\mathbf{x})$ (defined in Eq. (2)) denotes the global minimum solution set of the lower-level problem.
>
> We also agree with the reviewer that the gradient-based methods dynamics in the lower-level subproblem usually converge to local minimum (or stationary point) under non-convex setting. Generally, our algorithm converges to the optimized upper-level objective over lower-level stationary set. To address this issue, we add Assumption 3.1 (5) such that the true bilevel optimum attains over the lower-level stationary set (i.e., the local minimum mentioned by the reviewer). In summary, with  Assumption 3.1 (5), our algorithm converges to the true bilevel optimum under non-convex lower-level setting.

---

> > ### Comment · Reviewer_pFSa · 2021-08-11
> > **Response to Authors**
> >
> > Thank you for your clarifications. I acknowledge that I read other reviewers' comments and authors' responses. I modified my score accordingly as the authors addressed most of my concerns.

---

### Official Review · Reviewer_hf2w · 2021-07-16

**Rating:** 7
**Confidence:** 4

**Summary:**

This paper proposes a bilevel optimization method based on unrolled gradient-descent iterations of the lower-level (LL) problem, while providing a theoretical connection between the ideal bilevel optimization and the unrolled one when LL problem might not be convex. The main idea is two-fold: Regarding the initial point (i.e., z) of LL variable as another variable in the UL problem, and taking the pessimistic value from the unrolled iteration, i.e., max_k F(x, y_k(x, z) ). The theoretical derivation presents that the ideal bilevel problem (min phi(x)) is asymptotically equivalent to the unrolled version (min phi_K(x, z)). The resulting algorithm is very simple: (1) perform a few unrolled steps (with computational graph) for LL, (2) take the pessimistic estimate (y_k) from the steps, (3) and perform gradient descent for x and z for the UL problem based on the y_k(x, z). The paper also proposes a Nesterov version (for LL iterations). The proposed method shows state-of-the-art performance for a toy problem and two applications.

**Limitations And Societal Impact:**

Explaining the above limitation (the discrepancy) can be helpful for the completeness of the paper.

**Main Review:**

++ Relevant contribution for an important problem: Bilevel optimization with nonconvex objectives is popularly used nowadays. Solving this with unrolled iterations has been frequently used in the literature, but without any apparent theoretical guarantees. Providing this is definitely relevant.

-- Discrepancy between the theory and the algorithm: The most crucial problem I think is that the theory and the algorithm are describing different things. The convergence guarantee to global optima is true only because the inf operators in all the theoretical derivation are in fact global optimizers. However, the optimizer for the UL problem used in the proposed algorithm is actually a local one (projected gradient descent). For example, let us suppose the following case: F(x,y) = x + 1/2 y + cos(y), S(x) = argmin_y 1/2 y + cos(y), -1 <= x <= 1, -2pi <= y <= 2pi, and all the variables are scalars. In this case, if we pick a wrong z (and hyperparameters), then Algorithm 1 will converge to a non-global optimum. Moreover, the derivation says nothing about the convergence to local optima. This suggests that the proposed method excelling in the experiments might be not because of the guarantees provided in the paper. This is somewhat misleading and can confuse readers. I suggest the authors address this limitation in the main paper.

[After rebuttal]
My main concern has been well addressed in the authors' response. Providing the local solution convergence will be definitely helpful for the completeness. I'm satisfied with the response and I stay with my original score.

**Time Spent Reviewing:**

3

---

> ### Author Response · Authors · 2021-08-10
> **Response to Reviewer #1**
>
> We thank the reviewer for positive comments and insightful questions. Our response to the reviewer’s question is below.
>
> First of all, we would like to emphasize that, the mainstream of gradient-based bilevel methods in machine learning community (e.g., [1,2,3,4]) can be understood as a kind of sequential unconstrained minimization type scheme [5]. Specifically, in these papers, by unrolling the lower-level objectives with gradient-based schemes, one can obtain a series of unconstrained approximated subproblems. In fact, our work also falls into this category.
>
> As for the global optimum concern, we must clarify that this situation is indeed related to not only our IAPTT-GM, but all these mainstream gradient-based methods (i.e., sequential unconstrained minimization type scheme). In particular, the approximation quality of all these methods should be typically justified in theory from two aspects: If a series of global solutions of approximation subproblems are found, then a global solution of the original bilevel problem can be approximately achieved. Alternatively, if a series of local solutions of approximation subproblems are found, then a local solution of the original bilevel problem can be approximately achieved.
>
> In our submission paper, we actually construct a series of unconstrained approximated subproblems **under the weakest structure assumption among the literature, i.e., the lower-level non-convexity**. We have proved the approximation quality in our paper from the global solution aspect. To our best knowledge, we establish the first strict convergence guarantee for gradient-based method on bilevel with non-convex lower-level. For this purpose, technically, we propose IA and PTT, two new mechanisms to efficiently handle the lower-level non-convexity. IA actually paves the way for jointly optimizing both the UL variables and the dynamical initialization, while PTT adaptively reduces the complexity of backward recurrent propagation.
>
> In addition, within our theoretical approxiation framework, the local solution convergence can also be done by extending Lemma 3.2 to a local version. We will add the local solution convergence in the further version by deriving the proof arguments in a neighborhood sense.
>
> As for the particular numerical example mentioned by the reviewer, we have tested our IAPTT-GM and other mainstream gradient-based bilevel optimization methods (e.g., RHG and BDA) on it. Specifically, by conducting experiments with different initialization points for 500 times, we experimentally observe that IAPTT-GM can always converge to the global optima (i.e., $x = -1, y = -\frac{7}{6}\pi$) when $z$ is in the range $(-\frac{11}{6}\pi,\frac{1}{6}\pi)$. However, both RHG and BDA failed to obtain the true global optimal solution when their initialization point is sampled nearby the range boundary.
>
> [1] Luca Franceschi, Paolo Frasconi, Saverio Salzo, Riccardo Grazzi, and Massimiliano Pontil. "Bilevel programming for hyperparameter optimization and meta-learning." ICML, 2018.
>
> [2] Jordan Frecon, Saverio Salzo, and Massimiliano Pontil. "Bilevel learning of the group lasso structure." NeurIPS, 2018.
>
> [3] Junyi Li, Bin Gu, and Heng Huang. "Improved bilevel model: Fast and optimal algorithm with theoretical guarantee." arXiv:2009.00690 (2020).
>
> [4] Risheng Liu, Pan Mu, Xiaoming Yuan, Shangzhi Zeng, and Jin Zhang. "A generic descent aggregation framework for gradient-based bi-level optimization." arXiv:2102.07976 (2021)
>
> [5] Anthony V. Fiacco, and Garth P. McCormick. Nonlinear programming: sequential unconstrained minimization techniques. Society for Industrial and Applied Mathematics, 1990.

---

> > ### Comment · Reviewer_hf2w · 2021-08-20
> > **Response to the response**
> >
> > Thank you for the detailed response.
> >
> > I can understand the part about the mainstream in this area, and I would have no concern about this point if the authors add the local solution convergence they promised in the final version.
> >
> > About the toy problem, I realize that it was my absurd mistake. This is not an appropriate example and it should be more like F(x,y) = ( 1/2 x + cos(x) ) cos(y), S(x) = argmin_y cos(y), -2pi <= x <= 2pi, -2pi <= y <= 2pi. In this case, if the initial point is x^0 = -7/6 pi and z^0 = 0, then Algorithm 1 will only give a local optimum. However, I think that it suffices to just provide the local solution convergence in the final version.

---

> > > ### Author Response · Authors · 2021-08-21
> > > **Response to the comment from Ref #1**
> > >
> > > Thanks a lot for your comment. We deeply appreciate the toy example you present, which is a clear motivation for us to further study the local solution convergence. We will add the local solution convergence result in our final version for sure.

---

### Decision · Program_Chairs · 2021-09-27

**Decision:**

Accept (Spotlight)

**Comment:**

The paper proposes a truncated unrolling type method for solving a bi-level optimization with non-convex lower-level. It contains two interesting ideas, namely, Initialization Auxiliary and Pessimistic Trajectory Truncation. The proposed method admits a convergence guarantee, meanwhile, the computation cost is reduced. The techniques and analytic framework are novel, and enhance the understanding of gradient-based bi-level optimization methods. In summary, this work will possibly inspire some new bi-level optimization algorithms and technical analysis.